# Continual Learning in the Frequency Domain

**Ruiqi Liu**[1,2], **Boyu Diao**[1,2,†], **Libo Huang**[1], **Zijia An**[1,2], **Zhulin An**[1,2], **Yongjun Xu**[1,2]

[1]Institute of Computing Technology, Chinese Academy of Sciences, Beijing, China
[2]University of Chinese Academy of Sciences, Beijing, China

liuruiqi23@mails.ucas.ac.cn,
{diaoboyu2012, anzijia23p, anzhulin, xyj}@ict.ac.cn,
www.huanglibo@gmail.com

## Abstract

Continual learning (CL) is designed to learn new tasks while preserving existing knowledge. Replaying samples from earlier tasks has proven to be an effective method to mitigate the forgetting of previously acquired knowledge. However, the current research on the training efficiency of rehearsal-based methods is insufficient, which limits the practical application of CL systems in resource-limited scenarios. The human visual system (HVS) exhibits varying sensitivities to different frequency components, enabling the efficient elimination of visually redundant information. Inspired by HVS, we propose a novel framework called Continual Learning in the Frequency Domain (CLFD). To our knowledge, this is the first study to utilize frequency domain features to enhance the performance and efficiency of CL training on edge devices. For the input features of the feature extractor, CLFD employs wavelet transform to map the original input image into the frequency domain, thereby effectively reducing the size of input feature maps. Regarding the output features of the feature extractor, CLFD selectively utilizes output features for distinct classes for classification, thereby balancing the reusability and interference of output features based on the frequency domain similarity of the classes across various tasks. Optimizing only the input and output features of the feature extractor allows for seamless integration of CLFD with various rehearsal-based methods. Extensive experiments conducted in both cloud and edge environments demonstrate that CLFD consistently improves the performance of state-of-the-art (SOTA) methods in both precision and training efficiency. Specifically, CLFD can increase the accuracy of the SOTA CL method by up to 6.83% and reduce the training time by 2.6×. Code is available at https://github.com/EMLS-ICTCAS/CLFD.git

## 1 Introduction

Continual learning (CL) enables machine learning models to adjust to new data while preserving previous knowledge in dynamic environments [25]. Traditional training methods often underperform in CL because the adjustments to the parameters prioritize new information over old information, leading to what is commonly known as *catastrophic forgetting* [36]. While recent CL methods primarily concentrate on addressing the issue of forgetting, it is imperative to also consider learning efficiency when implementing CL applications on edge devices with constrained resources [37, 31], such as the NVIDIA Jetson Orin NX.

To mitigate *catastrophic forgetting*, a wide range of methods have been employed: *regularization-based* methods [49, 42, 30, 11, 3] constrain updates to essential parameters, minimizing the drift in network parameters that are crucial for addressing previous tasks; *architecture-based* methods [14, 24,

---

[†]Corresponding Author.

38th Conference on Neural Information Processing Systems (NeurIPS 2024).

35, 45, 52] allocate distinct parameters for each task or incorporate additional network components upon the arrival of new tasks to decouple task-specific knowledge; and *rehearsal-based* methods [2, 6, 8, 12, 10] effectively prevent forgetting by maintaining an episodic memory buffer and continuously replaying samples from previous tasks. Among these methods, rehearsal-based methods have been proven to be the most effective in mitigating *catastrophic forgetting* [6]. However, when the buffer size is constrained by memory limitations (e.g., on edge devices), accurately approximating the joint distribution using limited samples becomes challenging. Moreover, rehearsal-based methods often require frequent data retrieval from buffers. This process significantly increases both computational demands and memory usage, consequently limiting the practical application of rehearsal-based methods in resource-constrained environments.

By reducing the size of the input image, both the training FLOPs and peak memory usage can be significantly decreased, thereby enhancing the training efficiency of rehearsal-based methods. Concurrently, this method allows rehearsal-based methods to store more samples within the same buffer. However, directly downsampling the input image can significantly degrade the model's performance due to information loss. Owing to the natural smoothness of images, the human visual system (HVS) exhibits greater sensitivity to low-frequency components than to high-frequency components [48, 34], enabling the efficient elimination of visually redundant information. Inspired by HVS, we transfer the CL methods from the spatial domain to the frequency domain and reduce the size of input feature maps in the frequency domain. Several studies [48, 15, 13] have focused on accelerating model training in the frequency domain. However, two primary limitations hinder their direct application to CL: (1) These studies utilize Discrete Cosine Transform (DCT) to map images into the frequency domain, resulting in a complete loss of spatial information, which prevents the use of data augmentation techniques in rehearsal-based methods. (2) These studies introduce a significant number of cross-task learnable parameters, consequently increasing the risk of *catastrophic forgetting*.

To this end, we propose a novel framework called Continual Learning in the Frequency Domain (CLFD), which comprises two components: Frequency Domain Feature Encoder (FFE) and Class-aware Frequency Domain Feature Selection (CFFS). To reduce the size of input images, we propose the FFE. This method utilizes Discrete Wavelet Transform (DWT) to transform the original RGB image input into the frequency domain, thereby preserving both the frequency domain and spatial domain features of the image, which facilitates data augmentation. Furthermore, acknowledging that distinct tasks exhibit vary-

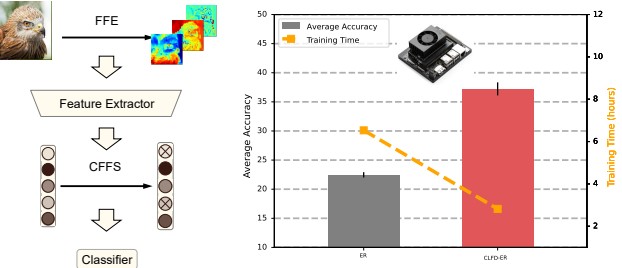

Figure 1: **Left**: Overview of CLFD. CLFD consists of two components: Frequency Domain Feature Encoder (FFE) and Class-aware Frequency Domain Feature Selection (CFFS). **Right**: On the NVIDIA Jetson Orin NX edge device, CLFD demonstrates a notable enhancement in both accuracy and efficiency compared to ER [4] on the split CIFAR-10 dataset.

ing sensitivities to different frequency components, we propose the CFFS method to balance the reusability and interference of frequency domain features. CFFS calculates the frequency domain similarity between inputs across different classes and selects distinct frequency domain features for classification. This method promotes the use of analogous frequency domain features for categorizing semantically similar inputs while concurrently striving to diminish the overlap of frequency domain features among inputs with divergent semantics. Our framework avoids introducing any cross-task learnable parameters, thereby reducing the risk of *catastrophic forgetting*. Simultaneously, by optimizing only the input and output features of the feature extractor, it facilitates the seamless integration of CLFD with various rehearsal-based methods. Figure 1 (right) demonstrates that CLFD significantly improves both the training efficiency and accuracy of rehearsal-based methods when implemented on the edge device.

In summary, our contributions are as follows:

- We propose the CLFD, a novel framework designed to improve the efficiency of CL. This framework enhances the training by mapping input features in the frequency domain and compressing these frequency domain features. To the best of our knowledge, this study represents the first attempt to

utilize frequency domain features to enhance the performance and the efficiency of CL on edge devices.

- CLFD stores and replays encoded feature maps instead of original images, thereby enhancing the efficiency of storage resource utilization. Concurrently, CLFD minimizes interference among frequency domain features, significantly boosting the performance of rehearsal-based methods across all benchmark datasets. These improvements can increase accuracy by up to 6.83% compared to the SOTA methods.

- We evaluate the CLFD framework on an actual edge device, showcasing its practical feasibility. The results indicate that our framework can achieve up to a $2.6\times$ improvement in training speed and a $3.0\times$ reduction in peak memory usage.

## 2 Related Work

### 2.1 Continual Learning

The current CL methods can be categorized into three primary types: *Regularization-based* methods [42, 30, 11, 22, 23] limit updates to key parameters to minimize drift in network parameters essential for previous tasks. *Architecture-based* methods [14, 24, 35, 45, 52] assign distinct parameters to each task or add network components for new tasks to decouple task-specific knowledge. *Rehearsal-based* methods [2, 6, 8, 12, 10] mitigate forgetting by maintaining an episodic memory buffer and continuously replaying samples from previous tasks to approximate the joint distribution of tasks during training. Among these, our framework focuses on rehearsal-based methods, as these methods are acknowledged as the most effective in mitigating *catastrophic forgetting* [6]. ER [38] enhances CL by integrating training samples from both the current and previous tasks. Expanding upon this concept, DER++ [6] enhances the learning process by retaining previous model output logits and utilizing a consistency loss during the model update. ER-ACE [7] safeguards learned representations and minimizes drastic adjustments required for adapting to new tasks, thereby mitigating *catastrophic forgetting*. Moreover, CLS-ER [4] mimics the interaction between rapid and prolonged learning processes by maintaining two supplementary semantic memories.

A limited number of works explore training efficiency in CL [46, 27, 16]. Among these methods, SparCL [46] reduces the FLOPs required for model training through the implementation of dynamic weight and gradient masks, along with selective sampling of crucial data. These methods accelerate the training process through pruning and sparse training. Nevertheless, our framework enhances efficiency by reducing the size of the input feature map, which is an orthogonal optimization to pruning.

### 2.2 Frequency domain learning

Some studies [48, 15, 18, 21] utilize DCT to map images into the frequency domain and enhance the inference speed of the models. However, these methods are not conducive to enhancing rehearsal-based methods. Previous research [6] indicates that data augmentation can significantly boost the performance of rehearsal-based methods. Nevertheless, utilizing DCT results in a total loss of spatial information, thereby restricting the application of data augmentation. Other studies [29, 33, 32, 47, 17, 13] employ DWT to improve the classification performance of models. While wavelet transform effectively preserves the spatial features of images, these methods are not well-suited for CL due to the substantial increase in learnable parameters they introduce. In CL, this proliferation of parameters significantly raises the risk of *catastrophic forgetting* across tasks. MgSvf [51] utilizes the frequency domain in the context of CL, focusing on the influence of different frequency components on model performance. In contrast, our framework delves into the differences in redundancy between the spatial and frequency domains. Compared to the spatial domain, CL in the frequency domain can more effectively remove redundant information from images, thereby improving the efficiency of CL.

## 3 Method

Our method, called Continual Learning in the Frequency Domain, is a unified framework that integrates two components: the Frequency Domain Feature Encoder, which transforms the initial RGB image inputs into the wavelet domain, and the Class-aware Frequency Domain Feature Selection,

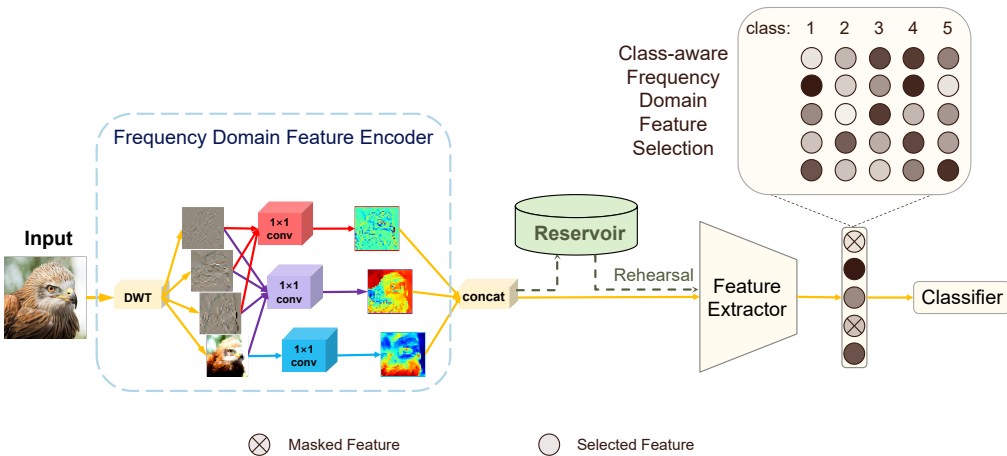

Figure 2: Illustration of the CLFD workflow. Initially, the original RGB image input is transformed into the wavelet domain through a Frequency Domain Feature Encoder. Subsequently, the feature extractor extracts the frequency domain features of these input feature maps. We propose a Class-aware Frequency Domain Feature Selection to selectively utilize specific frequency domain features, which are then inputted into the classifier for subsequent classification.

which balances the reusability and interference of frequency domain features. The entire framework is illustrated in Figure 2.

## 3.1 Problem Setting

The problem of CL involves sequentially learning $T$ tasks from a dataset, where in each task $t$ corresponds to a training set $\mathcal{D}^t = (x_i, y_i)_{i=1}^{N_t}$. Each task is characterized by a task-specific data distribution represented by the pairs $(x_i, y_i)$. To improve knowledge retention from previous tasks, we employ a fixed-size memory buffer, $\mathcal{M} = (x_i, y_i)_{i=1}^{\mathcal{B}}$, which stores data from tasks encountered earlier. Given the inherent limitations in CL, the model's storage capacity for past experiences is finite, thus $\mathcal{B} \ll N_t$. To address this constraint, we utilize reservoir sampling [44] to efficiently manage the memory buffer. In the simplest testing configuration, we assume that the identity of each upcoming test instance is known, a scenario defined as Task Incremental Learning (Task-IL). If the class subset of each sample remains unidentified during CL inference, the situation escalates to a more complex Class Incremental Learning (Class-IL) setting. This research primarily focuses on the more intricate Class-IL setting, while the performance of Task-IL is used solely for comparative analysis.

## 3.2 Discrete Wavelet Transform

DWT offers effective signal representation in both spatial and frequency domains [28], facilitating the reduction of input feature size. Compared with the DWT, DCT coefficients predominantly capture the global information of an image, but they fail to preserve the spatial continuity that is typical in normal images. In DCT, local spatial information is mixed, resulting in a loss of distinct local features. In contrast, DWT effectively integrates both spatial and frequency domain information, maintaining a balance between the two. Furthermore, the DWT method can be seamlessly integrated with data augmentation techniques in rehearsal-based methods, enhancing its applicability and effectiveness.

For 2D signal $X \in \mathbb{R}^{N \times N}$, The signal after DWT can be represented as:

$$X' = \begin{bmatrix} L \\ H \end{bmatrix} X \begin{bmatrix} L^T & H^T \end{bmatrix} = \begin{bmatrix} LXL^T & LXH^T \\ HXL^T & HXH^T \end{bmatrix} = \left[ \begin{array}{c|c} X_{ll} & X_{lh} \\ \hline X_{hl} & X_{hh} \end{array} \right], \qquad (1)$$

where $L$ and $H$ represent the low-frequency and high-frequency filters of orthogonal wavelets, respectively. These filters are truncated to the size of $\lfloor \frac{N}{2} \rfloor \times N$. The term $X_{ll}$ refers to the low-frequency component, while $X_{lh}, X_{hl}, X_{hh}$ represents the high-frequency components. We

select the Haar wavelet as the basis for the wavelet transform because of its superior computational efficiency [28], which is well-suited for our tasks.

## 3.3 Frequency Domain Feature Encoder

Previous methods [29, 47, 28] typically discarded high-frequency components $X_{lh}, X_{hl}, X_{hh}$ and retained low-frequency component $X_{ll}$. However, focusing solely on low-frequency component leaves many potentially useful frequency components unexplored. Low-frequency component compress the global topological information of an image at various levels, while high-frequency components reveal the image's structure and texture [20].

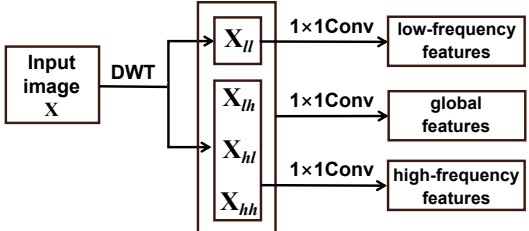

Figure 3: The Utilization of DWT in FFE.

Therefore, we employ three $1\times1$ point convolutions to integrate various frequency components. As shown in Figure 3, we use a $1\times1$ point convolution to merge low-frequency component $X_{ll}$ to obtain low-frequency features, another one to merge high-frequency components $X_{lh}, X_{hl}, X_{hh}$ to obtain high-frequency features, and a final one to merge all frequency components to obtain global features. These three merged features compose the input feature maps. By utilizing both low and high-frequency components in CL, we can better prevent the loss of critical information while reducing input feature size. Since each merged feature's width and height are half of the original image, another advantage of working in the frequency domain is that the spatial size of the original image (H × W × 3) is reduced by half in both width and height (H/2 × W/2 × 3) after FFE. With the reduced spatial size, the computational load and peak memory requirements of CL models decrease. Moreover, the reduction in spatial size means that more replay samples can be stored under the same storage resources, while also reducing the bandwidth required for accessing data. However, setting a specific frequency domain feature encoder for each task may result in significant *catastrophic forgetting*. Therefore, we freeze the FFE at the end of the first task's training.

## 3.4 Class-aware Frequency Domain Feature Selection

Considering that tasks are predominantly sensitive to specific frequency domain features extracted by a feature extractor, different tasks prioritize distinct frequency domain features. To this end, we propose the CFFS, designed to manage the issue of overlap in frequency domain features among samples from different classes. This method promotes comparable classes to utilize similar frequency domain features for classification, while also ensuring that samples from dissimilar classes employ divergent features. Consequently, this method reduces interference among various tasks and mitigates overfitting issues. For specific classes, we select a predetermined number of frequency domain features based on the absolute values of these features. Subsequently, unselected features are masked to prevent their interference in the classification process. We utilize a counter $\mathcal{F} \in \mathbb{R}^{C \times N}$ to track the number of selections for each frequency domain feature among samples associated with a specific class. $N$ and $C$ denote the dimensions and the classes of frequency domain features, respectively. We utilize cosine similarity to evaluate the frequency domain similarity between two class samples. To decrease computational complexity, only low-frequency component $X_{ll}$ is utilized for calculating cosine similarity. The similarity between class $i$ and class $j$ is expressed as follows:

$$S_{ij} = \frac{\mathbf{f}_i^\top \mathbf{f}_j}{\|\mathbf{f}_i\| \cdot \|\mathbf{f}_j\|}. \tag{2}$$

The value of cosine similarity is determined solely by the direction of the features, regardless of their magnitude. Consequently, $\mathbf{f}_i$ represents the sum of the low-frequency component of samples in class $i$. We then select the class that exhibits the greatest similarity and the class that displays the least similarity to the current class:

$$y_j^+ = \operatorname*{argmax}_{i \in \{1,...,K\}} S_{ij} \quad , \quad y_j^- = \operatorname*{argmin}_{i \in \{1,...,K\}} S_{ij}, \tag{3}$$

where $K$ represents the total number of classes in the preceding task. Several studies [39, 43] employ *Heterogeneous Dropout* [1] to enhance the selection of underutilized features for subsequent tasks. While this method helps manage the overlap of feature selection across different tasks, it overlooks

**Algorithm 1** Class-aware Frequency Domain Feature Selection Algorithm

---
**Input:** number of tasks $T$, training epochs of the $t$-th task $K_t$, dropout parameter $\lambda$ and $\beta_c$, frequency dropout epochs $\mathcal{E}$
**Initialize:** $P_f = 1, P_s = 1$
1: **for** $t = 1, \ldots, T$ **do**
2:      **for** $e = 1, \ldots, K_t$ **do**
3:          **if** $e < \mathcal{E}$ **then**
4:              **if** $t > 1$ **then**
5:                  Dropout features based on frequency dropout probabilities $1 - P_f$
6:                  **if** $e == 1$ **then**
7:                      Update $P_f$ at the end of the first epoch    (Eq. 4)
8:          **else**
9:              Dropout features based on semantic dropout probabilities $1 - P_s$
10:              Update $P_s$ at the end of each epoch    (Eq. 5)
11:          Select the top 60% of frequency domain features by response values for classification
12:          Update $\mathcal{F}$

---

similarities among classes. This oversight can negatively impact the effectiveness of selecting features in the frequency domain. To this end, we propose the *Frequency Dropout* method, which adjusts the probability of discarding frequency domain features based on class similarity. Specifically, let $[\mathcal{F}_y]_j$ denote the number of the j-th frequency domain feature when learning class $y$. The probability of selecting this feature in class $c$ while learning a new task is expressed as follows:

$$[P_f]_{c,j} = \lambda \exp\left(\frac{-[\mathcal{F}_{y_c^-}]_j}{\max_i[\mathcal{F}_{y_c^-}]_i} \cdot \alpha_c^-\right) + (1-\lambda)\left(1 - \exp\left(\frac{-[\mathcal{F}_{y_c^+}]_j}{\max_i[\mathcal{F}_{y_c^+}]_i} \cdot \alpha_c^+\right)\right),$$

$$\alpha_c^- = \frac{\overline{S}_c}{S_{cy_c^-}} \quad , \quad \alpha_c^+ = \frac{S_{cy_c^+}}{\overline{S}_c},$$

(4)

where $\overline{S}_c$ denotes the average cosine similarity between class $c$ and all classes in previous tasks. The parameters $\alpha_c^-$ and $\alpha_c^+$ control the intensity of the selection process. A higher value indicates a greater overlap of activated features with analogous classes and a reduced overlap with non-analogous classes. The coefficient $\lambda$ serves as a weighting factor that adjusts the selection of frequency domain features, determining whether the emphasis is more towards similar classes or less towards dissimilar ones. The *Frequency Dropout* probability is updated at the beginning of each task. After completing training for $\mathcal{E}$ epochs on a given task, *Semantic Dropout* [39] is employed instead of *Frequency Dropout*. It encourages the model to use the same set of frequency domain features for classification by setting the retention probability of frequency domain features in each class. This probability is proportional to the number of times that frequency domain feature has been selected in that class so far:

$$[P_s]_{c,j} = 1 - \exp\left(\frac{-[\mathcal{F}_c]_j}{\max_i[\mathcal{F}_c]_i}\beta_c\right),$$

(5)

where $\beta_c$ controls the strength of dropout. The probability of *Semantic Dropout* is updated at the end of each epoch, thereby enhancing the model's established frequency domain feature selection. This adjustment effectively regulates the extent of overlap in the utilization of frequency domain features. Algorithm 1 outlines the procedure for the CFFS.

## 4 Experiment

### 4.1 Experimental Setup

**Datasets.** We conduct comprehensive experimental analyses on extensively used public datasets, including Split CIFAR-10 (S-CIFAR-10) [6] and Split Tiny ImageNet (S-Tiny-ImageNet) [9]. The S-CIFAR-10 dataset is structured into five tasks, each encompassing two classes, while the S-Tiny-ImageNet dataset is divided into ten tasks, each comprising twenty classes. Additionally, the standard input image size for these datasets is $32 \times 32$ pixels.

Table 1: Comparison on different CL methods. CLFD consistently reduces the peak memory footprint of corresponding CL methods while simultaneously improving average accuracy. The highest results are marked in bold, and shadowed lines indicate the results from our framework.

| Buffer | Method | S-CIFAR-10 | | | S-Tiny-ImageNet | | |
|---|---|---|---|---|---|---|---|
| | | Class-IL | Task-IL | Mem | Class-IL | Task-IL | Mem |
| – | JOINT | $92.20_{\pm0.15}$ | $98.31_{\pm0.12}$ | - | $59.99_{\pm0.19}$ | $82.04_{\pm0.10}$ | - |
| | SGD | $19.62_{\pm0.05}$ | $61.02_{\pm3.33}$ | - | $7.92_{\pm0.26}$ | $18.31_{\pm0.68}$ | - |
| – | oEWC [41] | $19.49_{\pm0.12}$ | $68.29_{\pm3.92}$ | 530MB | $7.58_{\pm0.10}$ | $19.20_{\pm0.31}$ | 970MB |
| | SI [50] | $19.48_{\pm0.17}$ | $68.05_{\pm5.91}$ | 573MB | $6.58_{\pm0.31}$ | $36.32_{\pm0.13}$ | 1013MB |
| | LwF [30] | $19.61_{\pm0.05}$ | $63.29_{\pm2.35}$ | 316MB | $8.46_{\pm0.22}$ | $15.85_{\pm0.58}$ | 736MB |
| 50 | ER [38] | $29.42_{\pm3.53}$ | $86.36_{\pm1.43}$ | 497MB | $8.14_{\pm0.01}$ | $26.80_{\pm0.94}$ | 1333MB |
| | DER++ [6] | $42.15_{\pm7.07}$ | $83.51_{\pm2.48}$ | 646MB | $8.00_{\pm1.16}$ | $23.53_{\pm2.67}$ | 1889MB |
| | ER-ACE [7] | $40.96_{\pm6.00}$ | $85.78_{\pm2.78}$ | 502MB | $6.68_{\pm2.75}$ | $35.93_{\pm2.66}$ | 1314MB |
| | CLS-ER [4] | $45.91_{\pm2.93}$ | $89.71_{\pm1.87}$ | 1016MB | $11.09_{\pm11.52}$ | $40.76_{\pm9.17}$ | 3142MB |
| 50 | CLFD-ER | $45.56_{\pm3.71}$ | $84.45_{\pm0.85}$ | 205MB | $7.61_{\pm0.03}$ | $34.67_{\pm1.91}$ | 514MB |
| | CLFD-DER++ | $51.02_{\pm2.76}$ | $81.15_{\pm1.92}$ | 241MB | $10.69_{\pm0.27}$ | $31.55_{\pm0.39}$ | 658MB |
| | CLFD-ER-ACE | $\mathbf{52.74}_{\pm1.91}$ | $87.13_{\pm0.41}$ | 204MB | $10.71_{\pm2.91}$ | $38.05_{\pm11.98}$ | 514MB |
| | CLFD-CLS-ER | $50.13_{\pm3.67}$ | $85.30_{\pm1.01}$ | 401MB | $\mathbf{12.61}_{\pm0.95}$ | $37.80_{\pm3.08}$ | 1032MB |
| 125 | ER [38] | $38.49_{\pm1.68}$ | $89.12_{\pm0.92}$ | 497MB | $8.30_{\pm0.01}$ | $34.82_{\pm6.82}$ | 1333MB |
| | DER++ [6] | $53.09_{\pm3.43}$ | $88.34_{\pm1.05}$ | 646MB | $11.29_{\pm0.19}$ | $32.92_{\pm2.01}$ | 1889MB |
| | ER-ACE [7] | $56.12_{\pm2.12}$ | $90.49_{\pm0.58}$ | 502MB | $11.09_{\pm3.86}$ | $41.85_{\pm3.46}$ | 1314MB |
| | CLS-ER [4] | $53.57_{\pm2.73}$ | $90.75_{\pm2.76}$ | 1016MB | $16.35_{\pm4.61}$ | $46.11_{\pm7.69}$ | 3142MB |
| 125 | CLFD-ER | $55.76_{\pm1.85}$ | $88.29_{\pm0.16}$ | 205MB | $8.89_{\pm0.07}$ | $42.40_{\pm0.83}$ | 514MB |
| | CLFD-DER++ | $58.81_{\pm0.29}$ | $84.76_{\pm0.66}$ | 241MB | $15.42_{\pm0.37}$ | $40.94_{\pm1.30}$ | 658MB |
| | CLFD-ER-ACE | $58.68_{\pm0.66}$ | $89.35_{\pm0.34}$ | 204MB | $15.88_{\pm2.51}$ | $44.71_{\pm10.54}$ | 514MB |
| | CLFD-CLS-ER | $\mathbf{59.98}_{\pm1.38}$ | $87.09_{\pm0.43}$ | 401MB | $\mathbf{18.73}_{\pm0.91}$ | $49.75_{\pm2.01}$ | 1032MB |

**Evaluation metrics.** We use the average accuracy on all tasks to evaluate the performance of the final model:

$$ACC_t = \frac{1}{t}\sum_{\tau=1}^{t} R_{t,\tau} \tag{6}$$

We denote the classification accuracy on the $\tau$-th task after training on the t-th task as $R_{t,\tau}$. Moreover, we evaluate the training time, training FLOPs and peak memory footprint [46] to demonstrate the efficiency of each method. More experimental results can be found in Appendix F.

**Baselines.** We compare CLFD with several representative baseline methods, including three regularization-based methods: oEWC [41], SI [50] and LwF [30], as well as four rehearsal-based methods: ER [38], DER++ [6], ER-ACE [7] and CLS-ER [4]. In our evaluation, we incorporate two non-continual learning benchmarks: SGD as the lower bound and JOINT as the upper bound.

**Implementation Details** We expand the Mammoth CL repository in PyTorch [6]. For the S-CIFAR-10 and S-Tiny-ImageNet datasets, we utilize a standard ResNet18 [19] without pretraining as the baseline model, following the method outlined in DER++ [6]. All models are trained using the Stochastic Gradient Descent optimizer with a fixed batch size of 32. Additional details regarding other hyperparameters are detailed in Appendix D and E. For the S-Tiny-ImageNet dataset, models undergo training for 100 epochs, whereas for the S-CIFAR-10 dataset, training lasts for 50 epochs per task. In rehearsal-based methods, each training batch consists of an equal mix of new task samples and samples retrieved from the buffer. To ensure robustness, all experiments are conducted 10 times with different initializations, and the results are averaged across these runs.

## 4.2 Experimental Result

Table 1 presents a comparative analysis of the results on the S-CIFAR-10 and S-Tiny-ImageNet datasets, evaluated under Class-IL and Task-IL settings. The results elucidate that CLFD significantly

Table 2: **Ablation Study:** The Influence of systematically removing different components of CLFD-ER on model performance in S-CIFAR-10.

| Frequency Domain Feature Encoder | Class-aware Frequency Domain Feature Selection | Class-IL | Task-IL |
|:---:|:---:|:---:|:---:|
| ✗ | ✗ | $29.42_{\pm3.53}$ | $86.36_{\pm1.43}$ |
| ✓ | ✗ | $39.19_{\pm0.83}$ | $\mathbf{88.01}_{\pm0.06}$ |
| ✗ | ✓ | $37.80_{\pm5.78}$ | $85.78_{\pm2.43}$ |
| ✓ | ✓ | $\mathbf{45.56}_{\pm3.71}$ | $84.45_{\pm0.85}$ |

enhances the performance of various rehearsal-based CL methods. Specifically, the CLFD model has notably achieved SOTA accuracies across all buffer sizes in benchmark evaluations. By reducing the peak memory footprint by 2.4 ×, CLFD can augment the average accuracy of the ER method by up to 16.14%. Furthermore, when integrated with the SOTA method CLS-ER, CLFD can also increase its average accuracy by 6.41% and reduce its peak memory footprint by 2.5 ×. The superior performance of CLFD indicates that our proposed framework effectively mitigates *catastrophic forgetting* by improving the efficiency of storage resource utilization and minimizing interference among various frequency domain features. Moreover, the improvements implemented by CLFD across four different established rehearsal-based methods underscore its adaptability as a unified framework, highlighting its potential for integration with diverse CL methods.

### 4.3 Edge Device Results

We evaluate the acceleration performance of the CLFD utilizing the NVIDIA Ampere architecture GPU and Octa-core Arm CPU on the NVIDIA Jetson Orin NX 16GB platform. We measure the training time and accuracy of various methods using the S-CIFAR-10 dataset with a buffer size of 125. To expedite the training process, data augmentation techniques were omitted, resulting in accuracy results that vary from those reported in Table 1. Figure 4 illustrates the training time and average accuracy of various methods. When combined with various CL methods, CLFD significantly reduces training time while simultaneously enhancing model accuracy. By achieving approximately 2.4 × training acceleration, CLFD can attain the highest average accuracy of 47.64% when integrated with ER-ACE. This suggests that CLFD significantly enhances the efficiency of CL on edge devices by reducing the input feature map size.

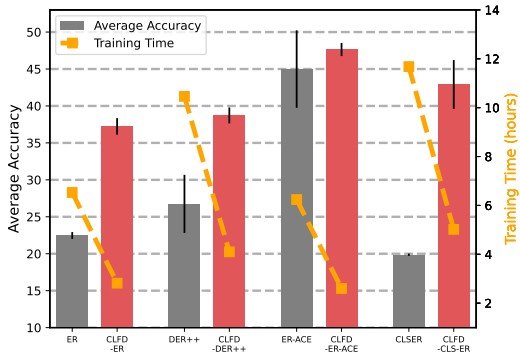

Figure 4: Comparison of different methods for S-CIFAR-10 dataset using Nvidia Jetson Orin NX with a buffer size of 125. When combined with various CL methods, CLFD significantly reduces training time while simultaneously enhancing model accuracy.

### 4.4 Ablation Study

In Table 2, we present a comprehensive ablation study of CLFD-ER employing a buffer size of 50 on the S-CIFAR-10 dataset. The results indicate that each component of our method makes a substantial contribution to enhancing accuracy. Comparing row 1 and 2, we can see that FFE enhances the utilization of storage resources by storing encoded features from the frequency domain rather than the original images. This method significantly improves the accuracy of rehearsal-based methods by optimizing the representation of stored data. Comparing rows 1 and 3, we can see that CFFS enhances the model's accuracy by mitigating the interference among frequency domain features across different tasks. CLFD achieves superior average accuracy by comprehensively integrating all components, thereby substantiating the efficacy of its individual elements.

Table 3: Comparison of CLFD and SparCL on S-CIFAR-10 dataset (Sparsity Ratio: 0.75, Buffer Size: 50).

| Method | S-CIFAR-10 | | |
|---|---|---|---|
| | Class-IL($\uparrow$) | Task-IL($\uparrow$) | FLOPs Train $\times 10^{15}(\downarrow)$ |
| ER [38] | 29.42$_{\pm 3.53}$ | 86.36$_{\pm 1.43}$ | 11.1 |
| SparCL-ER [46] | 43.74$_{\pm 2.91}$ | 85.01$_{\pm 3.86}$ | 2.0 |
| CLFD-ER | 45.56$_{\pm 3.71}$ | 84.45$_{\pm 0.85}$ | 2.8 |
| **CLFD-SparCL-ER** | **55.15$_{\pm 0.89}$** | **88.52$_{\pm 0.29}$** | **0.6** |

Table 4: Comparison of different frequency components.

| Method | S-CIFAR-10 | |
|---|---|---|
| | Class-IL($\uparrow$) | Task-IL($\uparrow$) |
| $X_{ll}$ | 49.59 | **83.92** |
| $X_{lh}$ | 41.77 | 79.92 |
| $X_{hl}$ | 44.45 | 82.76 |
| $X_{hh}$ | 34.19 | 74.27 |
| **FFE** | **51.02** | 81.15 |

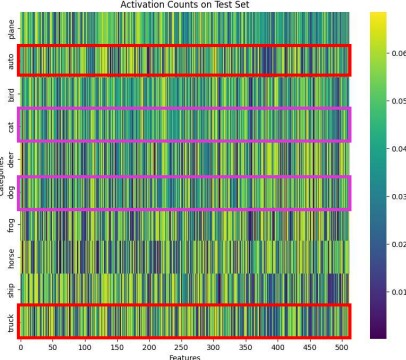

Figure 5: Frequency domain feature counts of the feature extractor trained on S-CIFAR10 with a buffer size of 125.

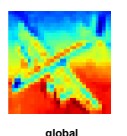

Figure 6: Visualization results of the FFE encoded image.

## 4.5 Model Analysis

In this section, we provide an in-depth analysis of CLFD.

**Sparse Training**  We compare our method with SparCL [46], a SOTA sparse training method. Table 3 presents the average accuracy and training FLOPs for CLFD and SparCL. Our method significantly enhances the average accuracy compared to SparCL. Although our method incurs higher training FLOPs than SparCL, this does not directly correlate with actual training speed. Our method accelerates training speed without requiring additional optimization. Conversely, pruning and sparse training methods that utilize masks often fail to translate into actual training time savings without optimizations at the compiler level. Furthermore, the combination of CLFD and SparCL not only achieves the highest accuracy but also leads to the lowest training FLOPs. The successful integration of CLFD and SparCL serves as an example of CLFD's adaptability to sparse training and pruning methods.

**The impact of different frequency components**  To explore the influence of different frequency components, we employ them as input feature maps and assess the CLFD-DER++ accuracy on the S-CIFAR-10 dataset under 50 buffer size. Table 4 presents the average accuracy across various frequency components. Utilizing the low-frequency components of images as input feature maps yields the highest accuracy among different frequency components. This suggests that CNN models demonstrate greater sensitivity to low-frequency channels compared to high-frequency channels. This finding aligns with the characteristics of the HVS, which also prioritizes low-frequency information. Despite this, FFE achieves the highest average accuracy, suggesting that high-frequency components are also significant. Optimal preservation of image information during downsampling is achieved only through the integration of components across different frequencies. Some examples of the encoded frequency domain feature maps are visualized in Figure 6.

**Frequency Domain Feature Selection**  To evaluate the effectiveness of frequency dropout in reducing interference among frequency domain features across diverse tasks, we calculate the

selection of specific frequency domain features for different classes. Specifically, we track the classification activities on the test set and conduct normalized counts of selections for the frequency domain features, as illustrated in Figure 5. We observe that feature selection patterns exhibit higher correlations among semantically similar classes. For instance, the classes "cat" and "dog" often select identical sets of features. Similarly, significant similarities in feature selection patterns are evident between "auto" and "truck". This result demonstrates the effectiveness of CFFS.

## 5 Conclusion

Inspired by the human visual system, we propose CLFD, a comprehensive framework designed to enhance the efficiency of CL training and augment the precision of rehearsal-based methods. To effectively reduce the size of feature maps and optimize feature reuse while minimizing interference across various tasks, we propose the Frequency Domain Feature Encoder and the Class-aware Frequency Domain Feature Selection. The FFE employs wavelet transform to convert input images into the frequency domain. Meanwhile, the CFFS selectively uses different frequency domain features for classification depending on the frequency domain similarity of classes. Extensive experiments conducted across various benchmark datasets and environments have validated the effectiveness of our method, which enhances the accuracy of the SOTA method by up to 6.83%. Moreover, it achieves up to a $2.6\times$ increase in training speed and a $3.0\times$ reduction in peak memory usage. We discuss the limitations and broader impacts of our method in Appendix A and B, respectively.

**Acknowledgement:** This work was supported by the Chinese Academy of Sciences Project for Young Scientists in Basic Research (YSBR-107) and the Beijing Natural Science Foundation (4244098).

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

## A    Limitations

One limitation of our framework is its focus on optimizing rehearsal-based methods. While the use of a rehearsal buffer throughout the CL process is widely accepted, there exist scenarios where rehearsal buffers are prohibited. Nonetheless, by reducing the size of the input feature map, our framework has the potential to accelerate various CL methods, though the implications for model performance require further investigation. Furthermore, our analysis was limited to scenarios with finite rehearsal buffer sizes, whereas many current studies focus on scenarios with infinite rehearsal buffer sizes. This shift is primarily due to memory constraints being less significant compared to computational costs. We will also continue to investigate the performance of our method in scenarios involving infinite rehearsal buffer sizes.

## B    Broader Impacts

Inspired by HVS, we propose a novel framework called CLFD, which utilizes frequency domain features to enhance the performance and efficiency of CL training. Its success has opened up opportunities for enhancing existing CL methods in the frequency domain. CLFD contributes to the responsible and ethical deployment of artificial intelligence technologies by improving CL performance and efficiency. This is accomplished through the efficient ability of models to update and refine their knowledge without the need for extensive retraining. This further facilitates the real-world application of CL.

Although CLFD serves as a comprehensive framework aimed at improving the efficiency of various CL methods, we must remain cognizant of its potential negative societal impacts. While CLFD improves model stability, it does so by compromising the expense of model plasticity, resulting in reduced accuracy when applied to new tasks. This trade-off requires specific consideration in applications where accuracy is crucial, such as healthcare [46]. Furthermore, as a powerful tool for enhancing the efficiency of CL methods, CLFD could also strengthen models designed for malicious applications [5]. Therefore, it is recommended that the community devise additional regulations to mitigate the malicious use of artificial intelligence.

## C    Dataset Licensing Information

- CIFAR-10 [26] is licensed under the MIT license.
- The licensing information for Tiny-ImageNet is unavailable. However, the dataset is accessible to researchers for non-commercial purposes.

## D    Hyperparameter Selection

Table A1 presents the selected optimal hyperparameter combinations for each method in the main paper. The hyperparameters include the learning rate (lr), batch size (bs), and minibatch size (mbs) for rehearsal-based methods. Other symbols correspond to specific methods. It should be noted that the batch size and minibatch size are held constant at 32 for all CL benchmarks.

## E    Experiment Details

### E.1    Experiment Platform

We conduct comprehensive experiments utilizing the NVIDIA GTX 2080Ti GPU paired with the Intel Xeon Gold 5217 CPU, as well as the NVIDIA Jetson Orin NX 16GB, boasting NVIDIA Ampere architecture GPU and Octa-core Arm CPU.

### E.2    Implementation Details

We set $\lambda = 0.5, \beta_c = 2$ in equation (4) and equation (5). We also set $\mathcal{E}$ at a value of 0.4 of the training epochs. In CFFS, we only select 60% of the frequency domain features for classification. We utilize the code from DER++ [6]. We extend our gratitude to the authors for their support and for providing

Table A1: Hyperparameters selected for our experiments.

| Method | Buffer | Split Tiny ImageNet | Buffer | Split CIFAR-10 |
|---|---|---|---|---|
| SGD | - | *lr:* 0.03 | - | *lr:* 0.1 |
| oEWC | - | *lr:* 0.03    *λ:* 90    *γ:* 1.0 | - | *lr:* 0.03    *λ:* 10    *γ:* 1.0 |
| SI | - | *lr:* 0.03    *c:* 1.0    *ξ:* 0.9 | - | *lr:* 0.03    *c:* 0.5    *ξ:* 1.0 |
| LwF | - | *lr:* 0.01    *α:* 1    *T:* 2.0 | - | *lr:* 0.03    *α:* 0.5    *T:* 2.0 |
| ER | 50 | *lr:* 0.1    *epoch:* 100 | 50 | *lr:* 0.1    *epoch:* 50 |
| | 125 | *lr:* 0.03    *epoch:* 100 | 125 | *lr:* 0.1    *epoch:* 50 |
| CLFD-ER | 50 | *lr:* 0.1    *epoch:* 100 | 50 | *lr:* 0.1    *epoch:* 50 |
| | 125 | *lr:* 0.03    *epoch:* 100 | 125 | *lr:* 0.1    *epoch:* 50 |
| DER++ | 50 | *lr:* 0.03    *α:* 0.1    *β:* 1.0    *epoch:* 100 | 50 | *lr:* 0.03    *α:* 0.1    *β:* 0.5    *epoch:* 50 |
| | 125 | *lr:* 0.03    *α:* 0.2    *β:* 0.5    *epoch:* 100 | 125 | *lr:* 0.03    *α:* 0.2    *β:* 0.5    *epoch:* 50 |
| CLFD-DER++ | 50 | *lr:* 0.03    *α:* 0.1    *β:* 0.5    *epoch:* 100 | 50 | *lr:* 0.03    *α:* 0.1    *β:* 0.5    *epoch:* 50 |
| | 125 | *lr:* 0.03    *α:* 0.1    *β:* 0.5    *epoch:* 100 | 125 | *lr:* 0.03    *α:* 0.2    *β:* 0.5    *epoch:* 50 |
| ER-ACE | 50 | *lr:* 0.03    *epoch:* 50 | 50 | *lr:* 0.03    *epoch:* 50 |
| | 125 | *lr:* 0.03    *epoch:* 50 | 125 | *lr:* 0.03    *epoch:* 50 |
| CLFD-ER-ACE | 50 | *lr:* 0.03    *epoch:* 50 | 50 | *lr:* 0.03    *epoch:* 50 |
| | 125 | *lr:* 0.03    *epoch:* 50 | 125 | *lr:* 0.03    *epoch:* 50 |
| CLS-ER | 50 | *lr:* 0.1    $r_S$: 0.04    $r_P$: 0.08    $α_S$: 0.999    $α_P$: 0.999    *λ:* 0.1    *epoch:* 50 | 50 | *lr:* 0.03    $r_S$: 0.05    $r_P$: 0.2    $α_S$: 0.999    $α_P$: 0.999    *λ:* 0.15    *epoch:* 50 |
| | 125 | *lr:* 0.1    $r_S$: 0.05    $r_P$: 0.08    $α_S$: 0.999    $α_P$: 0.999    *λ:* 0.1    *epoch:* 50 | 125 | *lr:* 0.03    $r_S$: 0.1    $r_P$: 0.9    $α_S$: 0.999    $α_P$: 0.999    *λ:* 0.15    *epoch:* 50 |
| CLFD-CLS-ER | 50 | *lr:* 0.1    $r_S$: 0.08    $r_P$: 0.16    $α_S$: 0.999    $α_P$: 0.999    *λ:* 0.15    *epoch:* 50 | 50 | *lr:* 0.03    $r_S$: 0.1    $r_P$: 0.5    $α_S$: 0.999    $α_P$: 0.999    *λ:* 0.15    *epoch:* 50 |
| | 125 | *lr:* 0.1    $r_S$: 0.1    $r_P$: 0.16    $α_S$: 0.999    $α_P$: 0.999    *λ:* 0.15    *epoch:* 50 | 125 | *lr:* 0.03    $r_S$: 0.2    $r_P$: 0.9    $α_S$: 0.999    $α_P$: 0.999    *λ:* 0.15    *epoch:* 50 |

the research community with the Mammoth framework, which facilitates a fair comparison of various CL methods under standardized experimental conditions. To ensure a fair comparison, we endeavor to closely align the experimental settings with those used in the Mammoth framework. However, we modified the data augmentation techniques within the Mammoth framework. The details of our data augmentation techniques are presented as follows.

### E.3   Data Augmentation

In line with [6], we employ random crops and horizontal flips as data augmentation techniques for both examples from the current task and the replay buffer. To ensure uniformity in data augmentation between the original images and the input features encoded with the FFE, random cropping is restricted to even pixels only.

## F   Additional Experiment Results

### F.1   Stability-Plasticity Trade-off

If the CL model can retain previously learned information, it is considered stable; if it can effectively acquire new information, it is considered plastic. To better understand how various methods balance

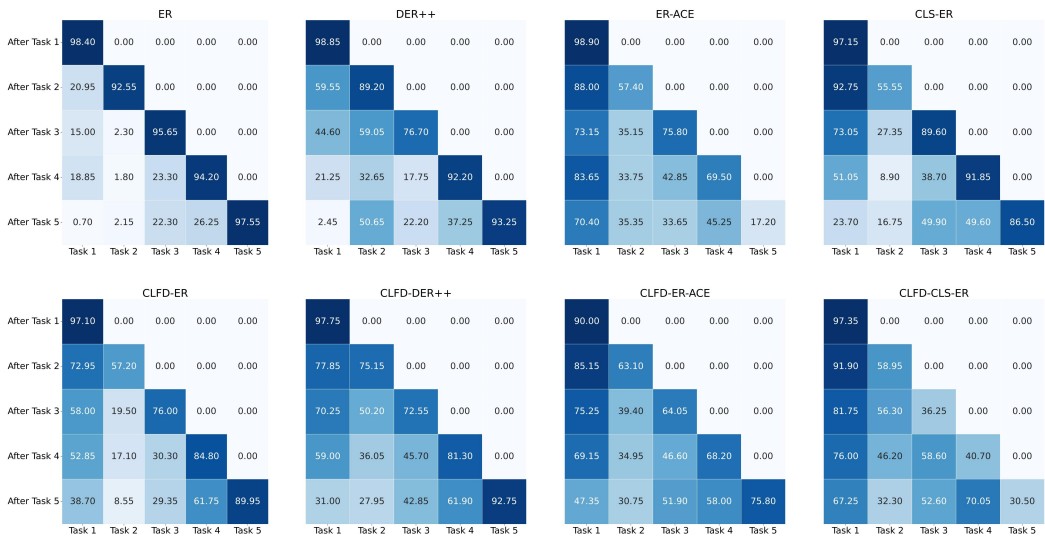

Figure 7: Performance of different methods by task. The heatmaps display the test set results for each task (x-axis) evaluated at the end of each sequential learning task (y-axis). We conducted experiments on the S-CIFAR-10 dataset using a buffer size of 50.

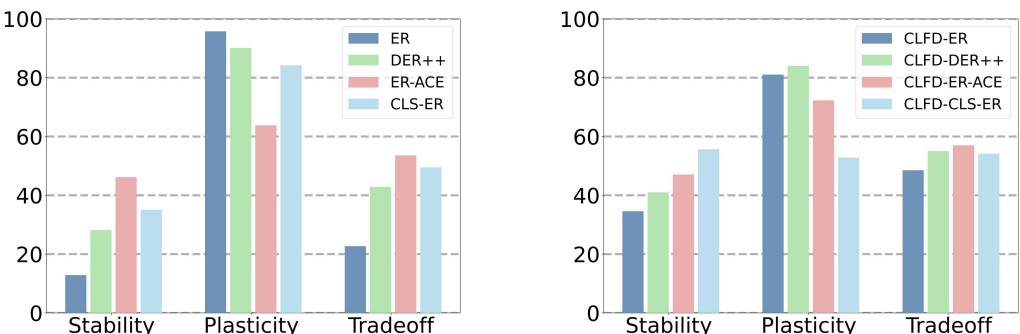

Figure 8: Stability-Plasticity Trade-off for CL models trained on S-CIFAR-10 under 50 buffer size.

stability and plasticity, we investigate the evolution of task performance as the model learns tasks sequentially. Figure 7 shows that CLFD consistently enhances the balance of all CL methods, delivering more uniform performance across all tasks. In addition, to further demonstrate the effectiveness of our method, we introduce a trade-off measure [40] that approximates how the model balances its stability and plasticity. Upon the model's completion of the final task $T$, its stability is evaluated by calculating the average performance across all preceding $T-1$ tasks as follows:

$$S = \frac{\sum_{\tau=1}^{T-1} R_{T,\tau}}{T-1} \tag{7}$$

The plasticity of the model (P) is evaluated by computing the average performance of each task after its initial learning i.e. the diagonal of the heatmap:

$$P = \frac{\sum_{\tau=1}^{T} R_{\tau,\tau}}{T} \tag{8}$$

Thus, the trade-off measure determines the optimal balance between the model's stability ($S$) and plasticity ($P$). This measure is calculated as the harmonic mean of $S$ and $P$.

$$\textit{Trade-off} = \frac{2SP}{S+P} \tag{9}$$

Figure 8 provides the stability-plasticity trade-off measure for different CL methods. ER and DER++ exhibit high plasticity, enabling them to rapidly adapt to new information. However, they lack the

Table A2: Forgetting results on S-CIFAR-10 dataset.

| Method | S-CIFAR-10 | | | |
| --- | --- | --- | --- | --- |
| | Class-IL($\downarrow$) | | Task-IL($\downarrow$) | |
| Buffer Size | 50 | 125 | 50 | 125 |
| ER [38] | $83.61_{\pm5.33}$ | $72.24_{\pm2.87}$ | $12.55_{\pm2.12}$ | $9.13_{\pm1.35}$ |
| DER++ [6] | $60.67_{\pm8.86}$ | $48.80_{\pm7.21}$ | $15.83_{\pm4.41}$ | $10.12_{\pm1.97}$ |
| ER-ACE [7] | $32.81_{\pm24.39}$ | $26.42_{\pm18.77}$ | $13.32_{\pm4.46}$ | $7.54_{\pm0.88}$ |
| CLS-ER [4] | $43.96_{\pm94.92}$ | $48.23_{\pm37.12}$ | $6.07_{\pm2.26}$ | $6.52_{\pm0.64}$ |
| CLFD-ER | $45.01_{\pm16.71}$ | $33.91_{\pm6.95}$ | $5.20_{\pm1.32}$ | $2.74_{\pm1.05}$ |
| CLFD-DER++ | $41.93_{\pm4.65}$ | $32.82_{\pm1.06}$ | $12.62_{\pm3.98}$ | $9.70_{\pm0.94}$ |
| CLFD-ER-ACE | $24.98_{\pm6.34}$ | $\mathbf{21.13}_{\pm1.45}$ | $4.92_{\pm2.19}$ | $\mathbf{3.04}_{\pm0.35}$ |
| CLFD-CLS-ER | $\mathbf{18.19}_{\pm2.19}$ | $22.20_{\pm2.90}$ | $\mathbf{4.22}_{\pm3.09}$ | $3.82_{\pm0.54}$ |

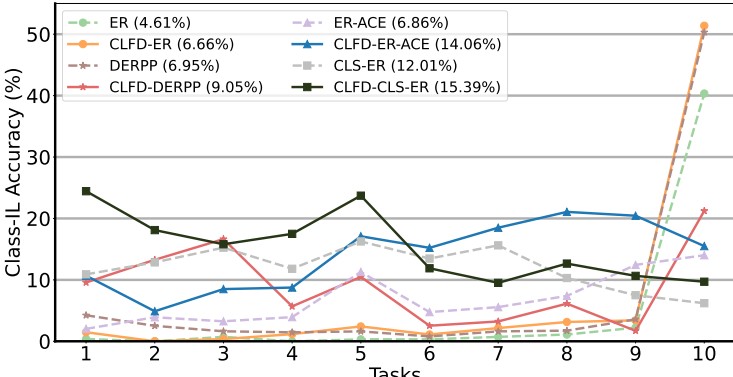

Figure 9: Comparison of Class-IL accuracy for different methods on the Split ImageNet-R dataset, divided into 10 tasks. The figure reports the accuracy of individual tasks at the end of CL training. The values in parentheses in the legend indicate the average accuracy.

ability to effectively retain previously acquired knowledge, leading to task recency bias issues. CLFD consistently improves the stability of CL methods, thereby reducing task reception bias and enhancing the balance between stability and plasticity.

## F.2 Forgetting

We use the Final Forgetting (FF) to measure the model's anti-forgetting performance:

$$FF_t = \frac{1}{t-1}\sum_{j=1}^{t-1} \max_{i\in\{1,...,t-1\}}(R_{i,j} - R_{t,j}), \tag{10}$$

A smaller FF value indicates that the model exhibits less forgetting of previous knowledge, thereby demonstrating stronger anti-forgetting performance. Table A2 presents the FF result. Our method consistently reduces forgetting in CL methods, demonstrating the effectiveness of CLFD in preserving prior knowledge rather than solely focusing on improving the accuracy of subsequent tasks.

## F.3 Split ImageNet-R Result

To further enhance the credibility of our results, we conduct additional experiments on ImageNet-R dataset. ImageNet-R, an extension of the ImageNet dataset, comprises 200 classes and a total of 30,000 images, with 20% designated for the test set. We divide ImageNet-R into ten tasks, each containing 20 classes. Input images are resized to $224 \times 224$ pixels. We test the Class-IL accuracy of each task with a buffer size of 500, maintaining consistent hyperparameters across all methods as those used in Split Tiny ImageNet. Figure 9 presents the experimental results, demonstrating that our method still significantly enhances the performance of various rehearsal-based CL methods,

Table A3: Task-IL and Class-IL accuracy under different feature selection proportions.

| Feature Selection Proportions | CLFD-ER | CLFD-ER-ACE |
|---|---|---|
| 10% | Task-IL: 85.67
Class-IL: 51.03 | Task-IL: 85.06
Class-IL: 50.37 |
| 30% | Task-IL: 84.91
Class-IL: 48.91 | Task-IL: 86.74
Class-IL: 50.64 |
| 50% | Task-IL: 83.88
Class-IL: 45.69 | Task-IL: 87.05
Class-IL: 52.12 |
| 90% | Task-IL: 87.97
Class-IL: 39.58 | Task-IL: 89.83
Class-IL: 54.20 |

even on more complex datasets. It is worth noting that the improvement in accuracy is not the sole advantage of our framework. By integrating our framework with rehearsal-based methods on the Split ImageNet-R dataset, training speed increased by up to $1.7\times$, and peak memory usage decreased by up to $2.5\times$. This demonstrates that our framework can significantly enhance the training efficiency of CL, thereby promoting its application on edge devices.

## F.4 Feature Selection Proportions in CFFS

We conduct additional ablation experiments to investigate the impact of feature selection proportions on model performance. We focus on two methods: CLFD-ER and CLFD-ER-ACE, as Figure 8 shows that ER is the most plastic method, while ER-ACE is the most stable. We conduct tests on the S-CIFAR-10 dataset with a buffer size of 50. Table A3 presents the result. In the Task-IL setting, both CLFD-ER and CLFD-ER-ACE achieve higher accuracy with higher feature selection proportions, as the inclusion of more features enables the model to better distinguish between classes within the tasks. It is important to highlight that in CLFD-ER, lower feature selection proportions can also lead to positive outcomes. This phenomenon can be attributed to the high plasticity of the CLFD-ER method, where reducing feature selection proportions aids in mitigating potential interference among various tasks. In the Class-IL setting, CLFD-ER achieves higher accuracy with lower feature selection proportions, while CLFD-ER-ACE exhibits superior performance with higher feature selection proportions. This observation suggests that for methods with high plasticity, we need to decrease the feature selection proportions to mitigate the overlap of frequency domain features, thereby minimizing the impact of new tasks on old tasks. Conversely, for methods with high stability, increasing the feature selection proportions allows us to utilize more frequency domain features to learn the current classes effectively.

