# OpenReview forum: "Continual Learning in the Frequency Domain"
_NeurIPS.cc/2024/Conference — NeurIPS 2024 poster_

### Official Review · Reviewer_KM8C · 2024-06-23

**Soundness:** 2
**Presentation:** 2
**Contribution:** 2
**Rating:** 5
**Confidence:** 3

**Summary:**

Inspired by the human visual system (HVS), this paper proposes a new framework called Continual Learning in the Frequency Domain (CLFD) for edge devices. For the input features of the feature extractor, CLFD employs wavelet transforms to map the original input image to the frequency domain, thereby reducing the size of the input feature map. In experiments on two public datasets, the performance of the proposed and conventional methods is discussed in terms of both accuracy and learning efficiency.

**Strengths:**

- Continuous learning in edge devices is a significant study from a practical point of view.
- The proposed method is simple and effective under limited conditions.
- A minimal survey of previous research is provided.

**Weaknesses:**

Throughout, the explanation of the proposed method needs to be more comprehensive. Experiments also need to be more comprehensive to demonstrate the effectiveness of the proposed method. Specifically, the paper has the following rooms for improvement.
- In line 60, there needs to be a clear explanation of why using frequency space is adequate. While it is interesting to get inspiration from HVS, there is no apparent reason why it is a means to the challenge of the proposed method (continuous learning on edge devices). In other words, the introduction needs a more logical structure.
- In Figure 2, the meaning of the symbols (e.g., \otimes, etc.) is unclear, so a specific explanation is needed. The clarity of the figure needs to be improved.
- In the description of the method, there is the following statement: "Considering that tasks are predominantly sensitive to specific frequency domain features extracted by a feature extractor, different To this end, we propose the CFFS, designed to manage the issue of overlap in frequency domain features among samples from different classes." However, no results from the analysis support this issue (fact). Furthermore, it needs to state why CFFS is the idea to solve this fact. Therefore, the design of the proposed method needs to be more convincing.
- Many grammatical errors need to be corrected. A comma or period is needed after the formula. (e.g., line 210: Where->where)
- The method's design throughout is very ad hoc and heuristic. For example, in Equation 4, no clear reason is given for the algorithm's design.
- Only the overall ACC represented by Equation 6 is evaluated in the experiment. In continuous learning, there are other evaluation measures (e.g., Forgetting measure). In addition, task-specific ACCs and other measures need to be evaluated. "[Chaudhry et al., 2018b] Arslan Chaudhry, Marc'Aurelio Ranzato, Marcus Rohrbach, and Mohamed Elhoseiny. Efficient lifelong learning with a-gem. In Proc. ICLR, 2018."
- In Table 1, there are no experiments for larger Buffer numbers. For example, previous studies (CLS-ER and SparCL-ER) have more buffers in their paper. Why are there no comparisons for larger numbers of buffers? It is unfair from the point of view of experiments in academic papers to publish only comparisons of conditions in which the proposed method is superior. (For example, edge devices that will be satisfied with larger buffer sizes may be developed.)
- Furthermore, it is difficult to judge the effectiveness of the proposed method since the experiments do not evaluate larger datasets; in this paper, only smaller datasets such as CIFAR and TinyImageNet are used.
- When the number of tasks increases, the proposed method will likely perform poorly for the latter tasks.

**Questions:**

The paper and the proposed method need both clarity and experimentation.
In particular, the proposed method's effectiveness is difficult to determine because of its ad hoc design and insufficient explanation.

- Please explain more about the necessity of utilizing frequency space for continuous learning for edge devices.
- Furthermore, I'm curious about the absence of experiments with larger Buffer numbers in Table 1. I think edge devices that will be satisfied with larger buffer sizes may be developed in the future.
- The overall ACC, represented by Equation 6, is the only evaluation. Other evaluation measures (e.g., forgetting measures) also exist in continuous learning. In addition, task-specific ACCs, etc., need to be evaluated. Please explain why you did not use these evaluation measures.

**Limitations:**

- For large data sets, the proposed method may need to be revised.
- When the number of buffers is large, the proposed method may be inferior to the conventional method.
- When the number of tasks increases, the proposed method is likely not to perform well for the latter tasks.

---

> ### Author Rebuttal · Authors · 2024-08-04
>
> Firstly, we extend our gratitude for the time and attention devoted to reviewing our paper. Below, we have carefully addressed each of your concerns to the best of our knowledge to improve the overall contribution of the paper.
>
> # The necessity of utilizing frequency space for continuous learning for edge devices.
>
> Thank you for this great question. Edge devices are constrained by limited memory and computing resources, underscoring the significance of training efficiency and peak memory usage in continual learning. Presently, continual learning primarily operates in the spatial domain, where the filters in a CNN are generally smooth due to the local pixel smoothness in natural images, resulting in spatial redundancies that impact the efficiency and memory usage of continual learning. Conversely, in the frequency domain, it is possible to train on images using distinct frequency domain features, significantly reducing redundancy in the learning process and improving the performance of continual learning methods.
>
> # The symbols in Figure 2 is unclear.
>
> Sorry for the unclear expression. $\otimes$ represent this feature being masked. In the final version of the paper, we will improve the clarity of the figure.
>
> # The design of the CFFS needs to be more convincing.
>
> In a previous study [1], it was found that continual learning has task-sensitive parameters for different tasks, and reducing the overlap of these parameters can significantly improve the performance of continual learning. Similarly, in the frequency domain, we observed that it also has task-sensitive parameters. As shown in Table 3, when we pruned 75% of the weights, the performance of the CLFD method improved significantly. Considering the presence of task-sensitive parameters in the classifier, which causes different tasks to be sensitive to specific frequency domain features, we propose CFFS to balance the reusability and interference of frequency domain features.
>
> [1] Sarfraz, Fahad, Elahe Arani, and Bahram Zonooz. "Sparse coding in a dual memory system for lifelong learning." AAAI. 2023.
>
> # Grammatical errors need to be corrected.
>
> We appreciate the reviewer's feedback regarding the grammatical errors. We will update the paper according to your suggestions in the final revision.
>
> # The method's design throughout is very ad hoc and heuristic.
>
> We agree with the reviewer that the method's design is very heuristic. However, the design of our method is well-founded. Heterogeneous Dropout has already been demonstrated to be effective in continual learning within the spatial domain [1,2]. Building on this foundation, we propose Frequency Dropout (Eq. 4). Additionally, in this work, we emphasize that our primary focus is on enhancing the efficiency of continual learning training by leveraging the frequency domain, thereby promoting its application on edge devices. Our research is among the first in this area, and we hope it will inspire the continual learning research community to further explore the frequency domain.
>
> [1] Sarfraz, Fahad, Elahe Arani, and Bahram Zonooz. "Sparse coding in a dual memory system for lifelong learning." AAAI. 2023.
>
> [2] Vijayan, Preetha et al. “TriRE: A Multi-Mechanism Learning Paradigm for Continual Knowledge Retention and Promotion.” NeurIPS. 2023.
>
> # Other evaluation measures.
>
> In Appendix Table A2, we provide the forgetting measure, while Figure 7 presents the task-specific ACCs, and Figure 8 illustrates the stability and plasticity of our framework. We will incorporate the key experimental results into the main text.
>
> # There are no experiments for larger Buffer numbers.
>
> We agree with the reviewer that conducting experiments on larger buffer sizes is essential. We conduct re-experiments under the buffer settings of SparCL-ER [1] and provide a detailed discussion in the general response.
>
> [1] Wang, Zifeng, et al. "Sparcl: Sparse continual learning on the edge." NeurIPS. 2022.
>
> # The experiments do not evaluate larger datasets.
>
> We agree with the reviewer's suggestion that incorporating larger datasets can enhance the credibility of the results. Therefore, we introduced the Split Imagenet-R dataset as an alternative and provided a detailed discussion in the general response.
>
> # The proposed method will likely perform poorly when the number of tasks increases.
>
> We sincerely thank the reviewers for raising this concern. On the Split Tiny ImageNet and Split ImageNet-R datasets, each comprising 10 tasks, our framework consistently demonstrates strong performance, further validating the effectiveness of our framework. We emphasize that although our accuracy remains high even with a large number of classes, the principal contribution of our work lies in introducing the concept of continual learning in the frequency domain, which significantly improves training efficiency and facilitates the application of continual learning on edge devices.
>
> We once again thank the reviewer for providing detailed feedback. We have made an utmost effort to resolve all the concerns raised. Please let us know in case we have missed something.

---

> > ### Comment · Reviewer_KM8C · 2024-08-10
> > **Official Comment by Reviewer KM8C**
> >
> > Thank you very much for your kind feedback. I deeply appreciate very thoughtful feedback. However, I still have a few concerns about the following;
> >
> >
> > 1) Scalability.:
> >
> > In the rebuttal, we discuss the validity of increasing the number of tasks with results of ten tasks. However, as noted in the paper below, in my understanding, the performance is also discussed with a larger number of tasks, e.g., 50 or 20 tasks, in a replay-based manner.
> >
> > - [i] Rebuffi, Sylvestre-Alvise, et al. "icarl: Incremental classifier and representation learning." in Proc. CVPR 2017. (https://arxiv.org/pdf/1611.07725)
> >
> > - [ii] Wu, Yue, et al. "Large scale incremental learning." in Proc.CVPR. 2019.(https://openaccess.thecvf.com/content_CVPR_2019/papers/Wu_Large_Scale_Incremental_Learning_CVPR_2019_paper.pdf)
> >
> >
> > 2) Positioning.:
> >
> > It is unclear whether the proposed method is "continual learning with the frequency domain (only) for edge devices" or "continual learning with the frequency domain (including for edge devices)". I am a bit confused about whether it is the former or the latter, as the explanation is inconsistent throughout the paper text, title, and rebuttal.

---

> > > ### Author Response · Authors · 2024-08-10
> > > **Reply to Reviewer KM8C**
> > >
> > > We thank the reviewer for taking the time to review our rebuttal. Below, we provide further clarification on the concerns you have raised.
> > >
> > > # Scalability.
> > >
> > > To validate the effectiveness of CLFD in addressing scalability issues, we divide the Split ImageNet-R dataset into 20 tasks and evaluate the performance of both ER-ACE and CLFD-ER-ACE. Additionally, we evaluate the average Class-IL accuracy after the completion of each task as the number of tasks increased from 10 to 20. The results are as follows:
> > >
> > > | Methods           | Class-IL Accuracy (%) | Task 10 | Task 11 | Task 12 | Task 13 | Task 14 | Task 15 | Task 16 | Task 17 | Task 18 | Task 19 | Task 20 |
> > > |-------------------|-----------------------|---------|---------|---------|---------|---------|---------|---------|---------|---------|---------|---------|
> > > | ER-ACE            | 10.63                  |    15.38     |     15.37    |    15.26     |    12.75     |    11.60     |    12.22     |    8.63     |    7.48     |    10.07     |    10.00     |    10.63     |
> > > | CLFD-ER-ACE       | 13.44                 |     15.93    |     16.43    |    17.20     |     13.59    |     15.42    |    13.08     |    9.75     |    14.69     |    13.26     |    12.79     |    13.44     |
> > >
> > >
> > > The results illustrate that our framework consistently enhances the performance of continual learning methods on the intricate Split ImageNet-R dataset, regardless of whether the dataset is divided into 10 tasks or 20 tasks. Furthermore, at any task boundary between 10 and 20, the accuracy of our framework consistently outperforms the baseline method. This indicates that our framework demonstrates strong scalability. We hope to provide results for additional methods before the end of the discussion period; however, time constraints and limited computational resources are significant challenges. In any case, we will incorporate these results into our final revision.
> > >
> > > # Positioning.
> > >
> > > We apologize for any confusion. Our framework is “continual learning with the frequency domain (including for edge devices).” The purpose of this framework is to leverage the frequency domain to enhance the performance and training efficiency of continual learning methods. Even when training in the cloud, our framework significantly reduces training time while improving accuracy. On edge devices, the advantages of our framework are even more pronounced due to the stricter memory constraints. Our framework can greatly increase training speed and reduce peak memory usage without requiring any additional optimization, an outcome that previous work has not achieved, thereby facilitating the deployment of continual learning on edge devices. We will revise the corresponding statements to avoid any potential confusion.
> > >
> > > We once again express our gratitude to the reviewer for the valuable feedback. Please let us know if any concerns persist. We would be more than willing to provide additional information to ensure a thorough understanding of our work.

---

> > > > ### Comment · Reviewer_KM8C · 2024-08-10
> > > >
> > > > Thank you very much for your thoughtful and meaningful discussion.
> > > >
> > > > ---
> > > > > Our framework is “continual learning with the frequency domain (including for edge devices).”
> > > >
> > > > OK.
> > > > I understand that the proposed framework is "continual learning with the frequency domain (including for edge devices)."  However, an algorithm that take advantage of frequency domains have already been proposed in the context of continual learning [a]. Also, exploiting frequency domains is one of the most common approaches in machine learning areas closely related to continual learning (e.g., domain generalization [b] and domain adaptation). We strongly recommend clarifying the position with such papers, e.g., in Sec. 2.3.
> > > >
> > > > [a] Zhao, Hanbin, et al. "Mgsvf: Multi-grained slow versus fast framework for few-shot class-incremental learning," IEEE TPAMI, 2021
> > > >
> > > > [b] Jiaxing Huang et al. "FSDR: Frequency Space Domain Randomization for Domain Generalization," in Proc. CVPR
> > > > , 2022
> > > >
> > > > ---
> > > > Please note that I believe that the authors' responses can solve most of my concerns about this paper, and I would like to re-rate this paper to 5: borderline accept.

---

> > > > > ### Author Response · Authors · 2024-08-11
> > > > > **Reply to Reviewer KM8C**
> > > > >
> > > > > We sincerely thank you for your thoughtful and highly constructive feedback, which has significantly contributed to improving the quality of our manuscript. Both FSDR [1] and Mgsvf [2] utilize the discrete cosine transform to incorporate frequency domain information, with a focus on how different frequency components influence the model. In contrast, our framework explores the differences in redundancy between the spatial and frequency domains. The frequency domain facilitates the more effective elimination of redundant information from images compared to the spatial domain, thereby enabling more efficient continual learning—an aspect not explored in previous work. Consequently, while Mgsvf primarily aims to enhance the model’s resistance to forgetting, our framework not only improves the accuracy of continual learning but also significantly increases training efficiency. We will expand the discussion on these two works in Section 3.2 to clarify our position. Once again, we sincerely appreciate your feedback in helping us to enhance the quality of our manuscript. If there are any other areas that require further clarification, we are ready to provide additional explanations as needed.
> > > > >
> > > > > [1] Jiaxing Huang et al. "FSDR: Frequency Space Domain Randomization for Domain Generalization," in Proc. CVPR , 2022
> > > > >
> > > > > [2] Zhao, Hanbin, et al. "Mgsvf: Multi-grained slow versus fast framework for few-shot class-incremental learning," IEEE TPAMI, 2021

---

### Official Review · Reviewer_wRV3 · 2024-07-03

**Soundness:** 2
**Presentation:** 2
**Contribution:** 3
**Rating:** 6
**Confidence:** 5

**Summary:**

Based on the research that human visual system (HVS) exhibits varying sensitivities to different frequency components, this paper proposes to do continual learning in the wavelet frequency domain to reduce the size of inputs. The proposed CLFD module includes feature extractor and feature encoder, where the feature encoder generate low-frequency features, global features and high-frequency features, and the feature extractor selects class-specific frequency features. The generated features are used to do continual learning.

**Strengths:**

1. This paper introduces the wavelet frequency domain features into CL. By encoding the low-frequency features and high-frequency features respectively, the proposed CLFD may have potential to mimic the human visual system for better learning results.
2. The feature extractor considers the class information of the frequency features to help the process of CL.

**Weaknesses:**

1. It is better to include larger databases such as imagenet-1k into the experiments to make the results more convincing.
2. Table-1 leaves some unclear items. For instance, the meaning of class-IL (class incremental learning) and task-IL (task incremental learning) should be explained. The meaning of shadowed lines and bold texts in the table should be included.
3. The classification performance of the proposed method seems have a large gap with the state-of-the-art methods on both cifar-10 and tiny-imagenet.

**Questions:**

1. How about the time-consuming of the processes of wavelet transform and the forward and backward of CLFD module?
2. Please clearly explain table-1 for better understanding.

**Limitations:**

1. The authors may consider to do some discuss about the different frequency domains such as fourier domain and discrete cosine domains.

---

> ### Author Rebuttal · Authors · 2024-08-04
>
> We extend our sincere gratitude for your thorough review of our paper. Below, we have diligently addressed each of your concerns to the best of our understanding, aiming to enhance the paper's overall contribution.
>
> # It is better to include larger databases.
>
> We agree with the reviewer that introducing larger databases can make the results more convincing. Therefore, we introduced the Split Imagenet-R dataset as an alternative and provided a detailed discussion in the general response.
>
> # Table-1 leaves some unclear items.
>
> We appreciate the reviewer's feedback regarding the shortcomings in Table 1. The highest results are marked in bold, and shadowed lines indicate the results from our framework. We introduced the definitions of Class-IL and Task-IL on line 147. In the simplest testing configuration, we assume that the identity of each upcoming test instance is known, a scenario defined as Task Incremental Learning (Task-IL). If the class subset of each sample remains unidentified during CL inference, the situation escalates to a more complex Class Incremental Learning (Class-IL) setting. We will update Figure 1 in the final version and provide definitions for Class-IL and Task-IL in the experiment.
>
> # The classification performance of the proposed method seems have a large gap with the state-of-the-art methods.
>
> The reason for the low classification performance is that we only use the polar buffer size. When we adjust the buffer size to be consistent with state-of-the-art methods, the classification performance significantly improves, and our method continues to enhance the performance of various rehearsal-based methods. We provide a detailed analysis in the general response.
>
> # The time-consuming of the processes of wavelet transform and CLFD module.
>
> We appreciate the reviewers for identifying the lack of training time analysis. We calculate the training time for the wavelet transform, CLFD module, ER method and CLFD-ER method throughout the entire Split CIFAR-10 training process. The wavelet transform accounts for only 1.5% of the ER training time, while the CLFD module accounts for only 13.1% of the ER training time. It is worth noting that when we integrated the CLFD module with the ER method, the training time was reduced by 2.3$\times$. This demonstrates the efficiency of our framework.
>
> | Methods | Training Time (s) |
> |---------|---------------------------------|
> | wavelet transform  |   127      |
> | CLFD     | 1125        |
> | ER     | 8542        |
> | CLFD-ER     | 3732        |
>
> # Some discuss about the different frequency domains.
>
> We discussed different frequency domains in sections 2.2 and 3.2. The discrete cosine transform and discrete Fourier transform both cause a complete loss of spatial information in images, making it difficult to utilize data augmentation techniques. Therefore, they are challenging to apply in continual learning.
>
> Once again, we express our gratitude for your thoughtful evaluation and consideration. We have conscientiously endeavored to address each of the concerns you raised, and we are dedicated to ensuring that our paper makes a meaningful contribution to the conference proceedings.

---

> ### Comment · Reviewer_wRV3 · 2024-08-09
>
> I thank the authors for answering my comments.
>
> I believe that the authors' responses can solve most of my concerns of this paper, and I would like to re-rate this paper to weak-accept.

---

### Official Review · Reviewer_2hDB · 2024-07-08

**Soundness:** 3
**Presentation:** 2
**Contribution:** 3
**Rating:** 6
**Confidence:** 3

**Summary:**

In this paper, the authors proposed a novel replay-based continual learning, which is named continual learning in the frequency domain (CLFD). The framework consists of two main modules, frequency domain feature encoder (FFE) and class-aware frequency domain feature selection (CFFS). FFE utilizes discrete wavelet transform (DWT) to transform the RGB images into a frequency domain. CFFS computes similarity and selects suitable frequency domain features for classification.

**Strengths:**

The strengths of this paper are listed below:
- The paper proposed a novel method that utilizes the frequency domain to decode the information of inputs.
- It can reduce the storage requirement to store a sample, which leads to less memory or more stored samples.
- They ran many experiments to show the performance.

**Weaknesses:**

The weaknesses of this paper are listed below:
- Some parts are not presented clearly. More details may included. Including a pseudocode may help.
- Some notations are not explained clearly, e.g. sec. 3.4.
- Even though there are many experiments, more ablation studies about hyperparameters may provide more information.

**Questions:**

How is the historical data stored? What format is the data in a memory buffer? How do you use it for replay?

Could you please give some more details about CFFS? The overall idea can be understood, but the details are unclear because of confusing notations.

**Limitations:**

As the authors have discussed, replay-based methods are unsuitable for all scenarios.

---

> ### Author Rebuttal · Authors · 2024-08-04
>
> We express our gratitude to the reviewer for dedicating time to thoroughly review our work. We value the positive feedback provided on our manuscript. Below, we address the weaknesses and queries raised:
>
> # More ablation studies about hyperparameters.
>
> We appreciate the feedback from the reviewers and acknowledge the importance of analyzing the robustness of hyperparameter selection. However, our framework can integrate with various continual learning methods, ablation studies on hyperparameters require substantial computational resources and extensive experimentation. Given this, we have demonstrated the effectiveness of our framework by using default hyperparameters for all datasets, methods, and buffer sizes.
>
> # The ways to replay data.
>
> We use the widely accepted continual learning repository Mammoth CL [1] for data replay. We adopt a reservoir sampling strategy [2] to save samples, storing all data in tensor format. During each batch’s training, we read an additional batch of samples from the memory buffer and train them together with the current batch.
>
> [1] Buzzega, Pietro, et al. "Dark experience for general continual learning: a strong, simple baseline." NeurIPS. 2020.
>
> [2] Vitter, Jeffrey S. "Random sampling with a reservoir." *ACM Transactions on Mathematical Software (TOMS)* 11.1 (1985): 37-57.
>
> # More details about CFFS.
>
> We apologize for any confusion. In the final version of the paper, we will optimize the details of CFFS. For a given task, we select only 60% of frequency domain features for classification. Each frequency domain feature is monitored for its selection frequency during training. Essentially, for each class, each frequency domain feature is assigned a selection counter $\mathcal{F}$ that increments when the feature's value is among the top 60%. Before selecting frequency domain features, each feature is assigned a dropout probability. During the initial epoch of each task training, we apply frequency dropout. In the later epochs of training, we use semantic dropout and update its probability after each epoch (Eq. 5). After completing the first task training, for subsequent tasks, we calculate the similarity of the classes (Eq. 2) and update the frequency dropout probability (Eq. 4) at the end of the first epoch. The pseudocode is provided in the attached PDF.
>
> We hope that the clarification provided has resolved your concerns and inquiries. Should you require further assistance or elaboration, we are willing to provide additional information to ensure a comprehensive understanding of our work.

---

> ### Author Response · Authors · 2024-08-11
> **More ablation studies about hyperparameters**
>
> We conduct supplementary ablation experiments to examine the influence of different feature selection proportions on model performance. Our analysis concentrated on two methods: CLFD-ER and CLFD-ER-ACE. Figure 8 in the appendix illustrates that ER exhibits the highest degree of plasticity, whereas ER-ACE demonstrates the greatest stability. We conduct tests on the S-CIFAR-10 dataset with a buffer size of 50. The accuracy results under the Task-IL setting are as follows:
>
> | Feature Selection Proportions | 10%  | 20%  | 30%  | 40%  | 50%  | 60%  | 70%  | 80%  | 90%  |
> |-------------------------------|------|------|------|------|------|------|------|------|------|
> | CLFD-ER                        |   85.67   |   85.47   |   84.91   |   84.89   |   83.88   |   84.45   |   84.49   |   84.99   |   87.97   |
> | CLFD-ER-ACE                    |   85.06   |   85.86   |   86.74   |   86.54   |   87.05   |   87.13   |   87.30   |   88.12   |   89.83   |
>
>
> And the accuracy results under the Class-IL setting are as follows:
>
> | Feature Selection Proportions | 10%  | 20%  | 30%  | 40%  | 50%  | 60%  | 70%  | 80%  | 90%  |
> |-------------------------------|------|------|------|------|------|------|------|------|------|
> | CLFD-ER                        |   51.03   |   49.97   |   48.91   |   46.67   |   45.69   |   45.56   |   43.88   |   41.79   |   39.58   |
> | CLFD-ER-ACE                    |   50.37   |   50.50   |   50.64   |   50.87   |   52.12   |   52.74   |   52.84   |  53.97    |   54.20   |
>
> We also conduct an ablation study on the other hyperparameters presented in Section E.2. Since our framework can be integrated with various continual learning methods, we do not use grid search; instead, we individually investigate the impact of each hyperparameter on continual learning performance. We focus on the CLFD-ER-ACE method for this study, as it performs well across all datasets and does not introduce any additional hyperparameters. We conduct tests on the S-CIFAR-10 dataset with a buffer size of 50. The accuracy results under the Class-IL setting are as follows:
>
> | $\beta_c$ in Eq. 5           | 0.5 | 1 | 1.5 | 2 | 2.5 | 3 | 3.5 | 4 | 4.5 | 5 |
> |-------------------------------|-----|---|-----|---|-----|---|-----|---|-----|---|
> | CLFD-ER-ACE                   | 50.70 | 50.19 | 51.47 | 52.74 | 50.29 | 52.48 | 51.65 | 51.41 | 50.73 | 52.36 |
>
> | $\lambda$ in Eq. 4           | 0.1 | 0.2 | 0.3 | 0.4 | 0.5 | 0.6 | 0.7 | 0.8 | 0.9 |
> |-------------------------------|-----|-----|-----|-----|-----|-----|-----|-----|-----|
> | CLFD-ER-ACE                   | 50.85 | 52.08 | 51.24 | 51.68 | 52.74 | 51.77 | 50.81 | 50.68 | 51.81 |
>
> | $\mathcal{E}$ in CFFS           | 5  | 10  | 15  | 20  | 25  | 30  | 35  | 40  | 45  |
> |---------------------------------|----|-----|-----|-----|-----|-----|-----|-----|-----|
> | CLFD-ER-ACE                     | 52.06 | 52.27 | 52.16 | 52.74 | 52.43 | 52.18 | 52.65 | 52.64 | 52.01 |
>
> Our choice of hyperparameters in the paper yields optimal performance, and our framework demonstrates robustness to these hyperparameter selections. We hope that these ablation studies will provide the reviewer a more thorough comprehension of our work. We extend our appreciation for the feedback provided. Should there be any aspects requiring further elucidation, we are prepared to offer further explanations.

---

> > ### Comment · Reviewer_2hDB · 2024-08-14
> >
> > Thank you for your response. I will keep the rating.

---

### Official Review · Reviewer_wmJG · 2024-07-11

**Soundness:** 3
**Presentation:** 2
**Contribution:** 2
**Rating:** 6
**Confidence:** 5

**Summary:**

This paper introduces a novel framework designed to enhance the efficiency and effectiveness of continual learning (CL) systems by leveraging frequency domain representations, inspired by the human visual system's varying sensitivity to different frequency components. This approach aims to address the limitations of existing rehearsal-based methods in CL, particularly under constraints like limited resources on edge devices. The framework, named Continual Learning in the Frequency Domain (CLFD), uses a wavelet transform to convert input images into frequency domain representations, optimizing both the input and output features of the feature extractor. This allows for better management of memory usage and computational demands, leading to improvements in both accuracy and training efficiency. Extensive experiments demonstrate that CLFD can enhance the performance of state-of-the-art rehearsal-based methods, achieving higher accuracy and reduced training times.

**Strengths:**

1.	The proposed method is inspired by the human visual system, which is sensitive to different frequency components and efficiently reduces visually redundant information. This represents a novel shift from traditional spatial domain methods to frequency domain methods in CL, suggesting a significant departure from established methods.
2.	CLFD significantly enhances the performance and training efficiency of continual learning systems on edge devices, achieving up to 6.83% higher accuracy and reducing training time by 2.6 times.
3.	The paper is structured to clearly present the problem of catastrophic forgetting in CL and how the proposed method addresses it by reducing the input feature map size and optimizing feature reuse. The methodology is described in detail, providing clarity on how the approach works and is implemented.
4.	By reducing training time and memory usage while improving accuracy, the framework can integrate with existing rehearsal-based methods without extensive modification, which underscores its practical significance and potential impact on the field.

**Weaknesses:**

1.	While the framework shows promising results in specific settings and datasets (like Split CIFAR-10 and Split Tiny ImageNet), the paper does not thoroughly discuss its performance across a wider range of scenarios or more complex datasets.
2.	The effectiveness of the proposed method is somewhat dependent on the buffer size used for rehearsal in continual learning. The reliance on buffer size might limit its utility in extremely constrained environments where memory is severely limited.
3.	Freezing the FFE based on the first task might introduce scalability issues, as the encoder might not efficiently handle the complexity introduced by a broader set of tasks or more diverse data.
4.	In Table 1, it is observed that under the Task Incremental Learning (Task-IL) setting, the integration of the CLFD framework leads to worsened performance in some experimental results. This highlights a potential weakness, as the paper does not sufficiently explain why the framework underperforms in these specific scenarios.
5.	The paper exhibits several technical and typographical issues that affects its formal presentation and readability. For example, the multiplication symbol in section 3.3 is represented as “x” instead of using a standard “$\times$”. In Eq. 2, symbols are not bolded. Figure 4’s caption lacks a period at the end. Other typos like writing “FFE” as “FFD” in the conclusion section, introduces confusion and can be misleading about key terms and components described in the paper. In addition, there are errors in the formulation of some equations. For instance, Eq. 7’s subscript should start from 1 instead of 0. Similarly, Eq. 8 is also incorrectly formulated.

**Questions:**

Please refer to the weaknesses.

**Limitations:**

The limitations have been discussed in Appendix A.

---

> ### Author Rebuttal · Authors · 2024-08-04
>
> We express our sincere gratitude to the reviewer for offering thoughtful feedback and providing a constructive evaluation of our work. The valuable input has significantly contributed to the improvement of our paper.
>
> # The paper does not thoroughly discuss its performance across a broader range of scenarios or more complex datasets.
>
> We agree with the reviewer that conducting experiments on a broader dataset is essential. Accordingly, we have supplemented our experiments with the Split Imagenet-R dataset and provided a detailed discussion in the general response.
>
> # The reliance on buffer size might limit its utility in extremely constrained environments where memory is severely limited.
>
> While the dependence on buffers does consume some storage resources, our method can achieve excellent performance even with smaller buffer sizes by preserving the frequency domain features of the image instead of the original image, as shown in Table 1. In contrast, many continual learning methods [1,2] that do not rely on buffers need extra parameters, such as teacher models, to prevent model forgetting, which use significantly more storage than rehearsal-based methods.
>
> More importantly, for continual learning on resource-constrained devices, the limitation of memory resources is much more critical than that of storage resources. This necessitates continual learning methods to have lower peak memory usage. As shown in Table 1, our method can achieve up to a 3.0 $\times$ reduction in peak memory usage. This effectively promotes the application of continual learning methods on resource-constrained devices.
>
> [1] Smith, James, et al. "Always be dreaming: A new approach for data-free class-incremental learning." *ICCV*. 2021.
>
> [2] Li, Zhizhong, and Derek Hoiem. "Learning without forgetting." *IEEE transactions on pattern analysis and machine intelligence* 40.12 (2017): 2935-2947.
>
> # Freezing the FFE based on the first task might introduce scalability issues.
>
> We agree with the reviewer that freezing the FFE does affect the plasticity of the model. However, not freezing the FFE leads to severe forgetting problems. To balance the plasticity and stability of the model, it is crucial to ensure that no cross-task learnable parameters are introduced. As shown in Figure 8 in the appendix, our method consistently improves the balance between plasticity and stability in continual learning methods.
>
> # The framework underperforms in the Task-IL setting.
>
> We appreciate the reviewers for identifying the lack of the Task-IL accuracy analysis. This result can be explained from two aspects: (1) FFE encodes the frequency domain features of the input image, reducing the size of the input feature map to one-fourth of its original size. This leads to fewer learnable features and increases the difficulty of classifying classes within the task. (2) To minimize the overlap of frequency domain features between inputs with different semantics across tasks, CFFS selects only a subset of frequency domain features for classification. This selection also increases the difficulty of classifying classes within tasks. However, as the task difficulty increases, the performance of our framework becomes more pronounced in the Task-IL setting. For instance, as shown in Table 1 in the attached PDF, our framework significantly enhances Task-IL accuracy on the Split Tiny ImageNet dataset with buffer sizes of 200 and 500.
>
> Considering that Task-IL is generally regarded as an easier CL scenario compared to Class-IL, this research, consistent with previous work [1], primarily focuses on the more intricate Class-IL setting (as mentioned in line 150). Furthermore, we emphasize that the novelty of this work lies not only in improving accuracy but also in significantly enhancing the training efficiency of continual learning. Additionally, we have validated our framework on edge devices, thereby promoting the application of continual learning in such environments.
>
> [1] Wang, Zifeng, et al. "Sparcl: Sparse continual learning on the edge." NeurIPS. 2022.
>
> # The technical and typographical issues.
>
> Thanks for pointing out these technical and typographical issues, we will revise them in the next release.
>
> We once again thank the reviewer for providing detailed and insightful feedback. Please let us know if there are any open points that we may have overlooked.

---

> > ### Comment · Reviewer_wmJG · 2024-08-09
> >
> > Overall, I am satisfied with the responses provided to the first and second weaknesses outlined in my initial review.
> >
> > However, regarding the third weakness, I noticed that there still lacks experimental evidence specifically validating that CLFD can effectively overcome scalability issues. This echoes the last weakness highlighted by Reviewer KM8C.
> >
> > Additionally, I am particularly concerned about the performance degradation observed in the results for the S-CIFAR-10 dataset under the Task-IL setting when CLFD is integrated. Considering that Task-IL is generally regarded as a simpler scenario compared to Class-IL, **the inferior performance in an easier setting raises doubts about the overall efficacy of the CLFD framework**. Moreover, when compared to SparCL [1], SCoMMER [2] and TriRE [3], the CLFD results still show an obvious gap. This discrepancy further accentuates my concerns regarding the effectiveness and robustness of CLFD.
> >
> > After considering the responses to the weaknesses I outlined and feedback from other reviewers, I maintain my original rating with a score of 4: borderline reject.
> >
> > [1] Wang, Zifeng, et al. "Sparcl: Sparse continual learning on the edge." NeurIPS. 2022.
> >
> > [2] Sarfraz, Fahad, Elahe Arani, and Bahram Zonooz. "Sparse coding in a dual memory system for lifelong learning." AAAI. 2023.
> >
> > [3] Vijayan, Preetha et al. “TriRE: A Multi-Mechanism Learning Paradigm for Continual Knowledge Retention and Promotion.” NeurIPS. 2023.

---

> > > ### Author Response · Authors · 2024-08-10
> > > **Reply to Reviewer wmJG**
> > >
> > > We thank the reviewer for swift response. Based on your suggestion, we conducted additional experiments.
> > >
> > > # CLFD can effectively overcome scalability issues.
> > >
> > > To validate the effectiveness of CLFD in overcoming scalability challenges, we divide the Split ImageNet-R dataset into 20 tasks and evaluate the performance of both ER-ACE and CLFD-ER-ACE. Additionally, we evaluate the average Class-IL accuracy after the completion of each task as the number of tasks increased from 10 to 20. The results are as follows:
> > >
> > > | Methods           | Class-IL Accuracy (%) | Task 10 | Task 11 | Task 12 | Task 13 | Task 14 | Task 15 | Task 16 | Task 17 | Task 18 | Task 19 | Task 20 |
> > > |-------------------|-----------------------|---------|---------|---------|---------|---------|---------|---------|---------|---------|---------|---------|
> > > | ER-ACE            | 10.63                  |    15.38     |     15.37    |    15.26     |    12.75     |    11.60     |    12.22     |    8.63     |    7.48     |    10.07     |    10.00     |    10.63     |
> > > | CLFD-ER-ACE       | 13.44                 |     15.93    |     16.43    |    17.20     |     13.59    |     15.42    |    13.08     |    9.75     |    14.69     |    13.26     |    12.79     |    13.44     |
> > >
> > > The results demonstrate that, on the complex Split ImageNet-R dataset, our framework consistently enhances the performance of continual learning methods, whether the dataset is divided into 10 tasks or 20 tasks. This indicates that our framework exhibits good scalability. We hope to provide results for additional methods before the end of the discussion period; however, time constraints and limited computational resources are significant challenges. In any case, we will incorporate these results into our final revision.
> > >
> > > # The performance of CLFD on the S-CIFAR-10 dataset under the Task-IL setting.
> > >
> > > Since we primarily focus on performance under the Class-IL setting, we select only 60% of the frequency domain features for classification in CFFS. When we select 90% of the frequency domain features, the accuracy results under the Task-IL setting with a buffer size of 50 are as follows:
> > >
> > > | Methods           | Task-IL Accuracy (%) |
> > > |-------------------|-------------------|
> > > | ER           | 86.36               |
> > > | DER++       | 83.51              |
> > > | ER-ACE       | 85.78              |
> > > | CLS-ER       | 89.71              |
> > > | CLFD-ER           | 87.97               |
> > > | CLFD-DER++       | 83.91              |
> > > | CLFD-ER-ACE       | 89.83              |
> > > | CLFD-CLS-ER       | 90.74              |
> > >
> > >
> > > The results indicate that our framework consistently improves accuracy under the Task-IL setting, which is also validated in Table 2 of the ablation study. When tested on the simpler S-CIFAR-10 dataset, removing the CFFS module enhances accuracy under the Task-IL setting, consistent with our previous explanations regarding CFFS. By adjusting the CFFS module, our framework is able to consistently improve accuracy under both Class-IL and Task-IL settings across all datasets.
> > >
> > > # Comparison with SCoMMER [1] and TriRE [2].
> > >
> > > Given that our framework efficiently integrates with various continual learning methods, we conduct integration tests of CLFD with SCoMMER [1] and TriRE [2]. We strictly maintain all hyperparameters and experimental settings consistent with those in the original papers. Using the Mammoth CL repository [3], we perform the tests on the S-CIFAR-10 dataset with a buffer size of 200 to ensure a fair comparison. The results are as follows:
> > >
> > > | Methods           | Class-IL Accuracy (%) |
> > > |-------------------|-------------------|
> > > | SCoMMER | 61.34               |
> > > | TriRE       | 56.18              |
> > > | CLFD-SCoMMER | 63.69               |
> > > | CLFD-TriRE       | 60.03              |
> > >
> > >
> > > The experimental results demonstrate that our framework consistently enhances the accuracy and training efficiency of both SCoMMER and TriRE. The observed variations in accuracy are attributed solely to the adjustments made to the data augmentation technique within the Mammoth CL repository, as a unified experimental setup is utilized. The original data augmentation technique from the repository cannot be directly applied to the frequency domain. Hence, we implement a simpler data augmentation technique, which is detailed in Sections E.2 and E.3 of the appendix.
> > >
> > > [1] Sarfraz, Fahad, Elahe Arani, and Bahram Zonooz. "Sparse coding in a dual memory system for lifelong learning." AAAI. 2023.
> > >
> > > [2] Vijayan, Preetha et al. “TriRE: A Multi-Mechanism Learning Paradigm for Continual Knowledge Retention and Promotion.” NeurIPS. 2023.
> > >
> > > [3] Buzzega, Pietro, et al. "Dark experience for general continual learning: a strong, simple baseline." NeurIPS. 2020.
> > >
> > > Once again, we express our gratitude to the reviewer for the valuable feedback provided. If there are any remaining concerns, please inform us. We are prepared to provide additional information to ensure a thorough understanding of our work.

---

> > > > ### Comment · Reviewer_wmJG · 2024-08-10
> > > >
> > > > Thank you for your timely and detailed response, which addresses most of my concerns. Given these clarifications, I am adjusting my rating to 5: borderline accept. However, I encourage the authors to include additional ablation studies that explore the effect of feature selection proportions on both Class-IL and Task-IL settings. It would also be beneficial to provide a deeper analysis of how these adjustments specifically impact performance across the different settings.

---

> > > > > ### Author Response · Authors · 2024-08-11
> > > > > **Reply to Reviewer wmJG**
> > > > >
> > > > > We sincerely appreciate the thoughtful feedback provided by the reviewer and the constructive evaluation of our work. Following your suggestion, we conduct additional ablation experiments to investigate the impact of feature selection proportions on model performance. We focus on two methods: CLFD-ER and CLFD-ER-ACE, as Figure 8 in the appendix shows that ER is the most plastic method, while ER-ACE is the most stable. We conduct tests on the S-CIFAR-10 dataset with a buffer size of 50. The accuracy results under the Task-IL setting are as follows:
> > > > >
> > > > > | Feature Selection Proportions | 10%  | 20%  | 30%  | 40%  | 50%  | 60%  | 70%  | 80%  | 90%  |
> > > > > |-------------------------------|------|------|------|------|------|------|------|------|------|
> > > > > | CLFD-ER                        |   85.67   |   85.47   |   84.91   |   84.89   |   83.88   |   84.45   |   84.49   |   84.99   |   87.97   |
> > > > > | CLFD-ER-ACE                    |   85.06   |   85.86   |   86.74   |   86.54   |   87.05   |   87.13   |   87.30   |   88.12   |   89.83   |
> > > > >
> > > > >
> > > > > And the accuracy results under the Class-IL setting are as follows:
> > > > >
> > > > > | Feature Selection Proportions | 10%  | 20%  | 30%  | 40%  | 50%  | 60%  | 70%  | 80%  | 90%  |
> > > > > |-------------------------------|------|------|------|------|------|------|------|------|------|
> > > > > | CLFD-ER                        |   51.03   |   49.97   |   48.91   |   46.67   |   45.69   |   45.56   |   43.88   |   41.79   |   39.58   |
> > > > > | CLFD-ER-ACE                    |   50.37   |   50.50   |   50.64   |   50.87   |   52.12   |   52.74   |   52.84   |  53.97    |   54.20   |
> > > > >
> > > > >
> > > > > In the Task-IL setting, both CLFD-ER and CLFD-ER-ACE achieve the highest accuracy with the highest feature selection proportions, which aligns with our previous analysis. It is noteworthy that in CLFD-ER, lower feature selection proportions can also yield favorable results. This is due to the high plasticity of the ER method, where reducing feature selection proportions helps to avoid interference between different tasks. In the Class-IL setting, CLFD-ER achieves higher accuracy with lower feature selection proportions, while CLFD-ER-ACE performs better with higher feature selection proportions. This phenomenon indicates that for methods with high plasticity, we need to lower the feature selection proportions to reduce the overlap of frequency domain features, thereby minimizing the impact of new tasks on old tasks. Conversely, for methods with high stability, increasing the feature selection proportions allows us to utilize more frequency domain features to learn the current classes effectively. We once again express our gratitude for your feedback, which will assist us in improving the quality of our manuscript. If there are any aspects that necessitate additional clarification, we are prepared to offer further explanations as needed.

---

> > > > > > ### Comment · Reviewer_wmJG · 2024-08-12
> > > > > >
> > > > > > I appreciate your timely response and insightful analysis. The authors have addressed most of the concerns I raised in my initial review. I recommend that the authors focus on resolving the remaining technical details that I initially mentioned, particularly ensuring all technical and typographical issues are thoroughly addressed. Additionally, further enhancement of the manuscript could be achieved by including detailed comparative and ablation studies about other hyperparameters as indicated in E.2. These additions would provide a more comprehensive understanding of the proposed method, and I believe that addressing these final points will make it a solid contribution to the field.

---

> ### Author Response · Authors · 2024-08-13
> **Supplementary experiments on the Split ImageNet-R dataset**
>
> We have completed the experiments with other methods on the Split ImageNet-R dataset under the 20-task setting. The results are as follows:
> | Methods           | Class-IL Accuracy (%) | Task 10 | Task 11 | Task 12 | Task 13 | Task 14 | Task 15 | Task 16 | Task 17 | Task 18 | Task 19 | Task 20 |
> |-------------------|-----------------------|---------|---------|---------|---------|---------|---------|---------|---------|---------|---------|---------|
> | ER                | 5.22                      |    9.49     |    8.82     |    7.44     |    6.87     |    5.93     |    5.34     |    4.70     |    5.63     |    5.13     |    5.33     |    5.22     |
> | CLFD-ER           | 5.23                      |    9.51     |    10.70     |    8.79     |    7.32     |    6.32     |    5.94     |    4.77     |    5.76     |    5.46     |   5.97      |    5.23     |
> | DER++             | 9.47                  |    16.37     |    15.23     |    9.58     |    12.15     |    11,31     |    10.64     |    9.79     |    9.78     |    9.66     |    6.10     |    9.47     |
> | CLFD-DER++        | 10.05                 |     18.04    |     17.64    |    12.16     |    15.66     |    12.56     |    12.13     |    11.00     |    10.48     |    10.17     |    7.86     |    10.05     |
> | ER-ACE            | 10.63                 |  15.38  |  15.37  |  15.26  |  12.75  |  11.60  |  12.22  |  8.63   |  7.48   |  10.07  |  10.00  |  10.63  |
> | CLFD-ER-ACE       | 13.44                 |  15.93  |  16.43  |  17.20  |  13.59  |  15.42  |  13.08  |  9.75   |  14.69  |  13.26  |  12.79  |  13.44  |
> | CLS-ER            | 8.72                      |    6.83     |    6.38     |    6.09     |    6.63     |    6.38     |    7.17     |    6.91     |    7.16     |    7.35     |    8.07     |    8.72     |
> | CLFD-CLS-ER       | 11.12                      |    9.77     |     10.30    |    9.99     |   10.64      |    10.72     |    11.32     |    11.14     |    11.22     |    11.36     |    11.43     |   11.12      |
>
> These results provide a more comprehensive demonstration of the scalability of our framework. We have included a comprehensive analysis of the framework's scalability in the final version of the paper.

---

> ### Author Response · Authors · 2024-08-13
> **Reply to Reviewer wmJG**
>
> We express our sincere appreciation for the thoughtful feedback offered by the reviewer and the constructive evaluation of our work. We have addressed all the technical details and typographical issues as per your suggestions. Additionally, following your recommendation, we conduct a detailed comparison and ablation study on the other hyperparameters presented in Section E.2. Since our framework can be integrated with various continual learning methods, we do not use grid search; instead, we individually investigate the impact of each hyperparameter on continual learning performance. We focus on the CLFD-ER-ACE method for this study, as it performs well across all datasets and does not introduce any additional hyperparameters. We conduct tests on the S-CIFAR-10 dataset with a buffer size of 50. The accuracy results under the Class-IL setting are as follows:
>
> | $\beta_c$ in Eq. 5           | 0.5 | 1 | 1.5 | 2 | 2.5 | 3 | 3.5 | 4 | 4.5 | 5 |
> |-------------------------------|-----|---|-----|---|-----|---|-----|---|-----|---|
> | CLFD-ER-ACE                   | 50.70 | 50.19 | 51.47 | 52.74 | 50.29 | 52.48 | 51.65 | 51.41 | 50.73 | 52.36 |
>
> | $\lambda$ in Eq. 4           | 0.1 | 0.2 | 0.3 | 0.4 | 0.5 | 0.6 | 0.7 | 0.8 | 0.9 |
> |-------------------------------|-----|-----|-----|-----|-----|-----|-----|-----|-----|
> | CLFD-ER-ACE                   | 50.85 | 52.08 | 51.24 | 51.68 | 52.74 | 51.77 | 50.81 | 50.68 | 51.81 |
>
>
> | $\mathcal{E}$ in CFFS           | 5  | 10  | 15  | 20  | 25  | 30  | 35  | 40  | 45  |
> |---------------------------------|----|-----|-----|-----|-----|-----|-----|-----|-----|
> | CLFD-ER-ACE                     | 52.06 | 52.27 | 52.16 | 52.74 | 52.43 | 52.18 | 52.65 | 52.64 | 52.01 |
>
>
> Our choice of hyperparameters in the paper yields optimal performance, and our framework demonstrates robustness to these hyperparameter selections. We hope that these ablation studies will provide the reviewer with a more comprehensive understanding of our work. We once again extend our gratitude for your thoughtful and highly constructive feedback.

---

> > ### Comment · Reviewer_wmJG · 2024-08-14
> >
> > Thank you for your timely response, which provides a more comprehensive understanding of the proposed method. Given these additional experiments, I have increased my rating to 6: Weak Accept. I also appreciate the authors' patient and detailed response, and I have no further questions.

---

### Author Rebuttal · Authors · 2024-08-04

We sincerely appreciate the reviewers' valuable comments and concerns. In the response below, we hope to have addressed all the points raised. Should there be any further questions or clarifications needed, please inform us so we can fully address any aspects of the paper during the rebuttal period. We plan to use additional pages to provide detailed clarifications on the issues raised by the reviewers, as outlined in the following responses. In this general response, we have clarified the reviewers' concerns regarding the accuracy of the proposed method with larger buffer sizes and on other datasets.

# The performance of our framework on larger buffer sizes.

We maintained the same buffer size selection as the state-of-the-art efficient continual learning method, SparCL [1], and conducted re-experiments on the Split CIFAR-10 and Split Tiny ImageNet datasets. Table 1 in the attached PDF presents the experimental results, demonstrating that our method still significantly enhances the performance of various rehearsal-based continual learning methods. It is important to note that while improving accuracy is a notable outcome, it is not the primary goal of this work. The novelty of this work lies in introducing the concept of continual learning in the frequency domain, which effectively reduces the training time and peak memory usage of continual learning methods, thereby facilitating their application on edge devices.

[1] Wang, Zifeng, et al. "Sparcl: Sparse continual learning on the edge." NeurIPS. 2022.

# The performance of our framework on other datasets.

We appreciate the reviewers' suggestion that incorporating larger datasets can enhance the credibility of our results. However, we consider Split Tiny ImageNet to be a substantial dataset. Numerous recent studies [1,2] have utilized Split Tiny ImageNet as the largest dataset in their experiments. Additionally, Mammoth CL [3], a widely adopted continual learning repository, also uses Split Tiny ImageNet as its largest dataset.

To further enhance the credibility of our results, we conducted additional experiments on Split ImageNet-R. We tested the Class-IL accuracy of each task with a buffer size of 500. Figure 1 in the attached PDF presents the experimental results, demonstrating that our method still significantly enhances the performance of various rehearsal-based continual learning methods, even on more complex datasets. It is worth noting that the improvement in accuracy is not the sole advantage of our framework. By integrating our framework with rehearsal-based methods on the Split ImageNet-R dataset, training speed increased by up to 1.7 $\times$, and peak memory usage decreased by up to 2.5 $\times$. This demonstrates that our framework can significantly enhance the training efficiency of continual learning, thereby promoting its application on edge devices.

[1] Gao, Qiang, et al. "Enhancing knowledge transfer for task incremental learning with data-free subnetwork." NeurIPS. 2023.

[2] Vijayan, Preetha et al. “TriRE: A Multi-Mechanism Learning Paradigm for Continual Knowledge Retention and Promotion.” NeurIPS. 2023.

[3] Buzzega, Pietro, et al. "Dark experience for general continual learning: a strong, simple baseline." NeurIPS. 2020.

---

### Decision · Program_Chairs · 2024-09-25

**Decision:**

Accept (poster)

**Comment:**

This work leverages the frequency domain to enhance the performance and training efficiency of rehearsal-based continual learning methods.  For the input features of the feature extractor, CLFD employs wavelet transform to map the original input image into the frequency domain, thereby effectively reducing the size of input feature maps. Regarding the output features of the feature extractor, CLFD selectively utilizes output features for distinct classes for classification, thereby balancing the reusability and interference of output features based on the frequency domain similarity of the classes across various tasks.

Initially the paper received mixed reviews. Reviewers have appreciation for the method design, extensive experiments, and writing, while raising concerns on lack of experiments on more complex datasets, method scalability, etc. The authors have provided a strong rebuttal that convinced the reviewer enough to support acceptance. The AC-panel concurs with their decision and strongly encourages the authors to include all additional clarifications and important experiments in the camera-ready version.